# E-MoFlow: Learning Egomotion and Optical Flow from Event Data via Implicit Regularization

**Wenpu Li**[1*]**, Bangyan Liao**[1,2*]**, Yi Zhou**[3]**, Qi Xu**[1,4]**, Pian Wan**[5]**,**
**Peidong Liu**[1†]**,**

[1]Westlake University    [2]Zhejiang University
[3]Hunan University    [4]Wuhan University    [5]Georgia Institute of Technology
**Project Page:** https://akawincent.github.io/EMoFlow/

## Abstract

The estimation of optical flow and 6-DoF ego-motion, two fundamental tasks in 3D vision, has typically been addressed independently. For neuromorphic vision (e.g., event cameras), however, the lack of robust data association makes solving the two problems separately an ill-posed challenge, especially in the absence of supervision via ground truth. Existing works mitigate this ill-posedness by either enforcing the smoothness of the flow field via an explicit variational regularizer or leveraging explicit structure-and-motion priors in the parametrization to improve event alignment. The former notably introduces bias in results and computational overhead, while the latter, which parametrizes the optical flow in terms of the scene depth and the camera motion, often converges to suboptimal local minima. To address these issues, we propose an unsupervised framework that jointly optimizes egomotion and optical flow via implicit spatial-temporal and geometric regularization. First, by modeling camera's egomotion as a continuous spline and optical flow as an implicit neural representation, our method inherently embeds spatial-temporal coherence through inductive biases. Second, we incorporate structure-and-motion priors through differential geometric constraints, bypassing explicit depth estimation while maintaining rigorous geometric consistency. As a result, our framework (called **E-MoFlow**) unifies egomotion and optical flow estimation via implicit regularization under a fully unsupervised paradigm. Experiments demonstrate its versatility to general 6-DoF motion scenarios, achieving state-of-the-art performance among unsupervised methods and competitive even with supervised approaches.

## 1   Introduction

Optical flow estimation [1] and 6-DoF camera motion recovery [2] are two core building blocks in many 3D vision tasks, playing a crucial role in providing motion and structural information for various downstream applications such as object tracking [3, 4], scene reconstruction [5, 6], and Simultaneous Localization and Mapping (SLAM) [7–9]. In classical computer vision, these two problems have been extensively studied and can be successfully solved independently, thanks to well-established feature extraction techniques [10] and data association [11] algorithms. However, when applied to the emerging field of neuromorphic vision [12], these traditional methods face significant challenges. The unique nature of event cameras [12], which reports asynchronous and sparse events instead of capturing frames, brings challenges in estimating optical flow and 6-DoF camera motion reliably. For example, optical flow estimation from event data faces the well-known aperture problem [13], where the learned flow is essentially the normal flow [14]. Furthermore, depth-free 6-DoF motion

---

*Equal Contribution: {liwenpu,liaobangyan}@westlake.edu.cn
† Corresponding author: liupeidong@westlake.edu.cn

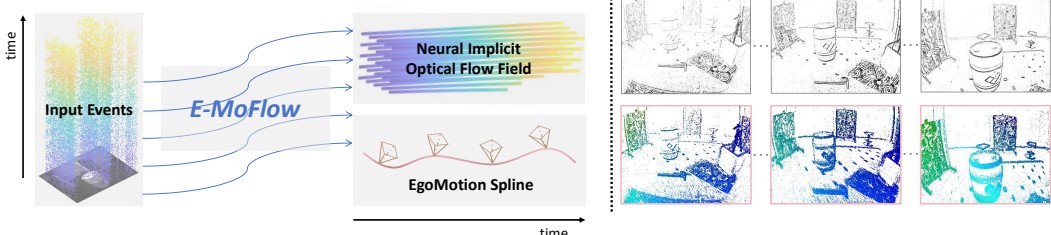

(a) Our method takes input of events, and predicts the dense and continuous optical flow and ego-motion.

(b) Recovered dense optical flow fields and corresponding images of warped events (IWE).

Figure 1: **Illustration of E-MoFlow**.

estimation in general scenes has been proved to be theoretically intractable [15] unless a locally constant depth assumption is imposed on the event data [16]. These challenges stem primarily from the absence of reliable long-term association in event data, rendering independent estimation of these two problems ill-posed and error-prone.

To overcome these challenges, existing approaches have primarily focused on regularization techniques to constrain the inherently ill-posed nature of these problems. One common strategy leverages spatial-temporal regularization to explicitly enforce optical flow continuity over both time and space [17–19]. These methods typically incorporate additional loss terms during optimization, which helps stabilize solutions but introduces trade-offs: the regularization constraints may bias flow estimation while simultaneously increasing computational complexity. In contrast, our approach embeds these regularization priors implicitly through learned representations. We formulate camera egomotion as a spline in the space of first-order kinematics [20] and optical flow as an implicit neural representation [21], which intrinsically encode spatial-temporal continuity. This representation-driven regularization eliminates the need for explicit constraint terms while maintaining solution stability.

Another line of works [16, 19] introduce geometric regularization by jointly estimating motion and depth. These approaches have shown improvements in optical flow accuracy, as depth information provides valuable constraints for motion estimation. However, these methods typically rely on motion fields [22] or re-projection equations [6] to relate optical flow, depth, and camera motion. This explicit depth estimation increases the degrees of freedom in the model, leading to a higher risk of local minima and instability during the optimization process. To address this issue, we adopt differential geometric constraints [23] to jointly estimate egomotion and optical flow without requiring explicit depth estimation. This approach implicitly incorporates geometric regularization, stabilizing the solution while retaining the ability to accurately estimate egomotion and flow.

In summary, **E-MoFlow** as shown in Fig. 1, unifies egomotion and optical flow estimation through implicit regularization under a fully unsupervised learning paradigm. We conduct extensive experiments across a variety of 6-DoF motion scenarios, demonstrating the applicability and robustness of our method. Experimental results demonstrate that **E-MoFlow** outperforms existing unsupervised methods and achieves comparable performance to supervised approaches.

## 2 Related Work

**Event-based Optical Flow Learning** Event cameras asynchronously measure per-pixel intensity changes, not absolute values at fixed intervals [12]. This enables high dynamic range and low-latency sensing. However, estimating optical flow from local event patches inherently suffers from the aperture problem [13, 14]. Because traditional optimization-based methods struggle to recover full motion fields from this ambiguous local data, learning-based approaches have become dominant. These learning-based optical flow methods are broadly categorized as supervised, semi-supervised, and unsupervised.

Supervised methods [24–32] rely on dense ground-truth optical flow for training, typically requiring large-scale synthetic or real-world datasets. However, acquiring accurate flow annotations at scale is prohibitively expensive, often leading to a significant sim-to-real gap that restricts their practical deployment in real-world environments. Semi-supervised methods, such as [18, 33, 34], mitigate this limitation by incorporating grayscale images and enforcing photometric consistency as a su-

pervisory signal. While these approaches reduce dependency on labeled data, their performance is highly sensitive to the quality of reconstructed intensity images, making them unreliable in extreme conditions (e.g., high-speed motion or low-light scenarios). In contrast, unsupervised learning methods [17, 19, 27, 32, 35–37] operate solely on event data, eliminating the need for external supervision. These methods typically optimize optical flow using contrast maximization (CMax) objectives [15], which align event warping with estimated motion. Depending on their learning paradigm, unsupervised approaches can be further divided into online and offline fashion. The former (e.g., [27, 32, 35–37]) employs neural networks to predict optical flow directly from event representations (e.g., event images or voxel grids). These approaches enable efficient, real-time inference but may suffer from error accumulation due to their feedforward nature. The latter (e.g., [17]) iteratively refines flow estimates from scratch for each new event batch. While more computationally intensive, this kind of method avoids the limitations of learned feature extraction, though often at the cost of reduced efficiency compared to online techniques.

In this work, we adopt an unsupervised learning paradigm that combines the efficiency of neural networks with high estimation accuracy. Our approach eliminates reliance on labeled data or auxiliary intensity images, ensuring robust performance across diverse and challenging scenarios.

**Event Camera Egomotion Estimation**   Direct egomotion estimation from event streams represents a fundamental yet highly challenging problem in event-based computer vision. Existing approaches primarily follow either 1) linear solver or 2) nonlinear optimization paradigms, each with distinct advantages and limitations.

Linear solver methods employ well-designed geometric constraints to derive closed-form motion solutions. For instance, EvLinearSolver [14] achieves 3-DoF rotation estimation and 6-DoF motion (by incorporating depth priors) through event-based normal flow constraints. Related work by [38, 39] utilizes event-based line features for translation estimation when angular velocity is known. As a result, these methods fundamentally require some form of prior knowledge, preventing complete egomotion recovery from event data alone. On the nonlinear optimization front, contrast-based approaches [15, 40–42] warp events to a reference timestamp and optimize motion parameters by maximizing the contrast in the resulting Image of Warped Events (IWE). While effective for certain motion patterns, these methods face critical challenges including degenerate solutions like event collapse [41] and inherent limitations in recovering full 6-DoF motion without depth priors (or necessarily assuming a constant depth shared by local events). Alternative spatial-temporal registration methods [43–46] leverage time surfaces for motion estimation, offering computational efficiency through sparse event processing but demonstrating increased sensitivity to noise compared to contrast-based techniques. More recently, [47] *et al.* propose an iterative optimization pipeline for 6-DoF motion estimation using event-based line features.

In conclusion, current solutions - whether linear or nonlinear - still cannot achieve reliable 6-DoF motion estimation in general scenarios without restrictive assumptions. This fundamental limitation highlights the need for more robust approaches capable of handling the full complexity of egomotion estimation from event data.

**Flow and Motion Joint Estimation**   Joint estimation of optical flow and camera motion typically builds upon the motion field theory [22] or re-projection constraints [6], which inherently require depth estimation. [19] pioneers an event-based framework using discretized spatial-temporal volumes to jointly predict optical flow and ego-motion, employing motion compensation through motion blur based and temporal re-projection losses for unsupervised learning. Similarly, [48] utilizes the Evenly-Cascaded Convolutional Network to parameterize depth and pose, employing the re-projection error between warped event images and the current frame as a supervisory signal to jointly optimize depth, motion, and optical flow.

A distinct approach is presented by [16], which introduces a contrast maximization framework for simultaneous estimation of optical flow, depth, and egomotion through motion field parameterization. However, these joint estimation methods face a critical challenge: the inclusion of depth estimation expands the parameterization space, introducing additional degrees of freedom that require stronger regularization to avoid convergence to local minima.

**Neural Flow Fields**   Recent studies [21, 49] have demonstrated the significant potential of using deep neural network as continuous and memory-efficient implicit representations for tasks such as view synthesis [50] [51], scene flow estimation [52, 53], and Non-Rigid Structure from Motion [54].

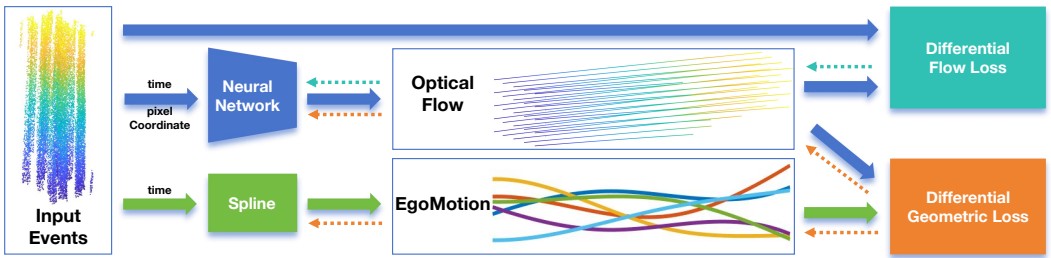

Figure 2: **Training Pipeline of E-MoFlow.** Given the input event data, we use differential flow loss Eq. (8) and differential geometric loss Eq. (9) to train the neural network Eq. (2) and spline parameters Eq. (6). These two losses are optimized together until convergence. Solid arrows denote the forward process; dashed arrows denote gradient backpropagation.

This class of methods leverages coordinate-based networks to store the mapping from spatio-temporal coordinates to target vectors, such as radiance values or motion fields. The inherent smoothness bias of the neural network architecture itself serves as a powerful "neural prior," providing implicit regularization for downstream problems and replacing traditional handcrafted regularizers.

By modeling the differential properties of the scene [52–54], this paradigm can be extended to represent entire temporal dynamics as Ordinary Differential Equations (ODEs) and to estimate complex, long-term 3D trajectories. This provides a solid foundation for our proposed method.

## 3 Methodology

In this section, we will first introduce the continuous representations of optical flow and camera motion (Sec. 3.1). Next, we will introduce the differential losses used in our work (Sec. 3.2), including differential flow loss Eq. (8) and differential geometric loss Eq. (9). Finally, we summarize our training pipeline in Sec. 3.3.

### 3.1 Continuous Representations

To include the spatial-temporal consistency prior through continuous representations implicitly, we model ego-motion and optical flow as spline [20] and implicit neural representations [21], respectively.

**Continuous Flow.** Implicit neural representations [21] model the continuous signal through a spatial-temporal coordinate based neural network. Specifically, given a time $t$ and a normalized pixel coordinate $\mathbf{x}$, the neural network will output a normalized optical flow vector $\mathbf{u}_\theta(t, \mathbf{x})$. We can write it as

$$\mathbf{u}_\theta(t, \mathbf{x}) = \text{NN}_\theta(t, \mathbf{x}), \tag{1}$$

where we denote the parameters of this neural network as $\theta$. The detailed implementation can be found in Sec. 4.1. Besides, following the Neural Ordinary Differential Equation (Neural ODE) [55], we can reformulate the warping trajectory of an event $e_k \doteq \{\mathbf{x}_k, t_k\}$ as a neural ODE solution. In particular, given the initial condition, the warping terminal point at time $t$ can be denoted as $e_k(t) \doteq \{\mathbf{x}_k(t), t\}$ and it satisfies the following equation:

$$\frac{d\mathbf{x}_k(t)}{dt} = \text{NN}_\theta(t, \mathbf{x}_k(t)), \quad \mathbf{x}_k(t_k) = \mathbf{x}_k. \tag{2}$$

This ODE can be integrated by any off-the-shelf ODE solvers (e.g., euler [56], rk4 [57], dopri5 [58]), with the solution defining the event warping trajectory:

$$\mathbf{x}_k(t) = \mathbf{x}_k + \int_{t_k}^{t} \text{NN}_\theta(s, \mathbf{x}_k(s)) \, ds \tag{3}$$

For the backward gradient, this can be solved by a reverse adjoint ODE. Given a scalar-valued loss function at the reference time point $L(\mathbf{x}_k(t_{\text{ref}}))$, the adjoint state $\mathbf{a}_k(t) = dL/d\mathbf{x}_k(t)$ namely the gradient at time $t$ can be represented by

$$\frac{d\mathbf{a}_k(t)}{dt} = -\mathbf{a}_k(t)^\top \frac{\partial \text{NN}_\theta(t, \mathbf{x}_k(t))}{\partial \mathbf{x}_k(t)}, \quad \mathbf{a}_k(t_{\text{ref}}) = \frac{dL(\mathbf{x}_k(t_{\text{ref}}))}{d\mathbf{x}_k(t_{\text{ref}})}. \tag{4}$$

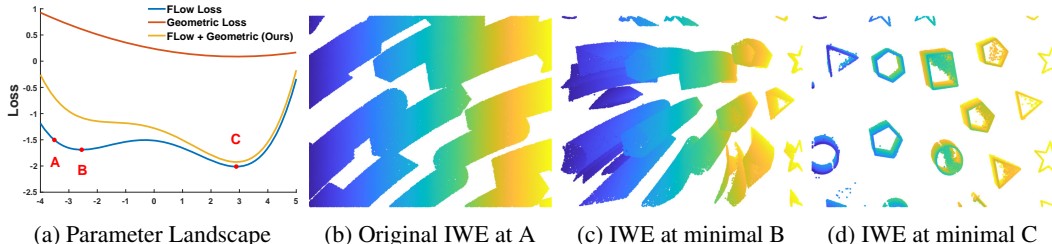

(a) Parameter Landscape     (b) Original IWE at A     (c) IWE at minimal B     (d) IWE at minimal C

Figure 3: **Landscape of different loss functions.** By jointly estimating these two losses, we can avoid getting stuck in local minima, making it possible to solve the ill-posed problem.

Finally, to get the gradient to the parameters, we can perform a simple integral along time as:

$$\frac{dL}{d\theta} = -\int_{t_{\text{ref}}}^{t_k} \mathbf{a}_k(t)^\top \frac{\partial \text{NN}_\theta(t, \mathbf{x}_k(t))}{\partial \theta} dt \tag{5}$$

*Remark.* Unlike previous work, which directly models displacement [17, 19], our approach directly models optical flow (velocity field). This enables our method to be applicable to scenes with more aggressive motion. Additionally, direct backpropagation of gradients can lead to gradient explosion and excessive memory usage [55]. The Neural ODE [55] approach mitigates this by modeling the backpropagation of gradients as an adjoint ODE, significantly reducing memory consumption.

**Continuous Motion.** Unlike optical flow, camera ego-motion is low-dimensional, and we represent it using a cubic B-spline [20]. Specifically, given $n + 1$ control points $\beta_i \in \mathbb{R}^6$, the angular and linear velocities $\boldsymbol{\omega}_\beta(t), \boldsymbol{\nu}_\beta(t)$ at time $t$ can be derived as follows:

$$[\boldsymbol{\omega}_\beta(t); \boldsymbol{\nu}_\beta(t)] = \sum_{i=0}^{n} \mathbf{B}_{i,3}(t)\beta_i, \tag{6}$$

where $\mathbf{B}_{i,3}$ denotes the cubic basis function of B-spline. For more details, please refer to the supplementary material.

### 3.2 Loss Functions

Building on the continuous representations introduced earlier, in this section, we continue by presenting the differential losses used in our work. Specifically, the differential flow loss Eq. (8) employs the CMax loss [15] to learn optical flow. Additionally, a differential geometric loss is proposed in Eq. (9) to handle the 6-dof motion scenario. By combining these two losses, we are able to overcome the ill-posed problem while simultaneously avoiding the need for depth estimation.

**Differential Flow Loss** For the differential flow loss, we follow the standard contrast maximization pipeline [15] to learn the continuous optical flow field. Given a set of events input $\mathcal{E} = \{e_k\}_{k=1}^{N}, e_k \doteq \{\mathbf{x}_k, t_k\}$ with size $N$, following the definition of warping trajectory defined in Eq. (3), we can warp each of them to a reference time $t_{\text{ref}}$, denoted as $\mathcal{E}(t_{\text{ref}}) = \{e_k(t_{\text{ref}})\}_{k=1}^{N}$. Then, all the event are accumulated into an image of warped events (IWE),

$$I(\mathbf{x}, \mathcal{E}(t_{\text{ref}})) = \sum_{k=1}^{N} \mathcal{N}(x; \mathbf{x}_k(t_{\text{ref}}), \sigma^2), \tag{7}$$

where $\sigma$ defines the gaussian smoothing kernel size. Then, by the following differential flow loss, we can measure the concentration of events.

$$L_{\text{flow}}(\mathcal{E}(t_{\text{ref}}), \theta) = -\frac{1}{HW} \sum_{i=1}^{H} \sum_{j=1}^{W} (I_{ij} - \mu_I)^2, \tag{8}$$

where $H, W$ and $\mu_I = \sum_{i=1}^{H} \sum_{j=1}^{W} I_{ij}$ denotes the hight, width and the mean of IWE, respectively.

**Differential Geometric Loss**  In the multi-view geometry literature, there are two equations connect optical flow and ego-motion: the motion field equation [22] and the epipolar equation [23]. Although not widely known, the differential epipolar equation [23] can theoretically be viewed as the differential forms of the epipolar constraint. Our differential geometric loss is primarily based on this equation. Specifically, the 6-DOF motion differential geometric loss in homogeneous coordinate can be defined to be:

$$L_{\text{geometry}}(t, \mathbf{x}, \theta, \beta) = \left\| \hat{\mathbf{u}}_\theta(t, \mathbf{x})^\top [\boldsymbol{\nu}_\beta(t)]_\times \hat{\mathbf{x}} - \hat{\mathbf{x}}^\top \mathbf{s}_\beta(t) \hat{\mathbf{x}} \right\|_2^2, \tag{9}$$

where $\mathbf{s}_\beta(t) = \frac{1}{2} \left( [\boldsymbol{\nu}_\beta(t)]_\times [\boldsymbol{\omega}_\beta(t)]_\times + [\boldsymbol{\omega}_\beta(t)]_\times [\boldsymbol{\nu}_\beta(t)]_\times \right)$, hat $\hat{\cdot}$ denotes the homogeneous coordinate and $[\cdot]_\times$ denotes the skew-symmetric operation.

*Remark.* As shown in Fig. 3, by jointly estimating these two losses, we can prevent optical flow estimation from getting trapped in local minima, making it possible to solve the inherently ill-posed problem involving translational motion.

### 3.3  Training Pipeline

After collecting the representations and losses, we are ready to build the unsupervised training pipeline. As shown in Fig. 2, given a sequence of event data $\mathcal{E}$, we use differential flow and differential geometric losses to train the neural network and spline parameters.

$$\min_{\theta, \beta} \mathbb{E}_{t_{\text{ref}}} \left[ L_{\text{flow}}(\mathcal{E}_{\text{neigh}}(t_{\text{ref}}), \theta) \right] + \mathbb{E}_{\{\mathbf{x}, t\} \sim \mathcal{E}} \left[ L_{\text{geometry}}(t, \mathbf{x}, \theta, \beta) \right], \tag{10}$$

For the differential flow loss, we randomly select a reference time $t_{\text{ref}}$ and a set of surrounding events $\mathcal{E}_{\text{neigh}}(t_{\text{ref}})$ around it, then evaluate the differential flow loss $L_{\text{flow}}(\mathcal{E}_{\text{neigh}}(t_{\text{ref}}), \theta)$ and back propagate the gradient using Neural ODE to train the network parameters $\theta$. For the differential geometric loss, we randomly sample some events $e = \{\mathbf{x}, t\} \sim \mathcal{E}$, collect the corresponding optical flow for each event $\text{NN}_\theta(t, \mathbf{x})$ and camera velocity $[\boldsymbol{\omega}_\beta(t); \boldsymbol{\nu}_\beta(t)]$ derived from the spline $\beta$ at time $t$, compute and accumulate the differential geometric loss, and then back propagate the gradient to train both the neural network $\theta$ and the spline $\beta$. These two losses are optimized simultaneously until convergence.

## 4  Experiments

### 4.1  Experimental Setup

**Datasets and Metrics.**  We conduct comprehensive evaluations on the MVSEC dataset [64], which is the de facto standard dataset used in prior work to benchmark optical flow and 6-DoF ego-motion estimations. This dataset contains both indoor sequences recorded by drones and outdoor sequences recorded by vehicles. Additionally, we benchmarked optical flow estimation on the more challenging DSEC [65] dataset, which features complex textures, diverse motion patterns, and varying lighting conditions. For optical flow evaluation, we compute three standard metrics: endpoint error (EPE), angular error (AE) and the percentage of pixels with EPE $> 3$ pixels (denoted by "% Out"), exclusively over valid ground truth regions with event activity in the evaluation interval. For motion estimation accuracy, we adopt the root mean square error (RMSE) metric proposed in [14] to measure angular velocity $(°/s)$ and linear velocity $(m/s)$ errors.

**Baselines.**  Optical flow estimation using event camera is a widely explored task with various methodological paradigms. We categorize existing approaches into four primary classes: Supervised Learning(SL), which requires ground truch optical flow for training (e.g., E-RAFT [26], EV-FlowNet-EST [25], EV-FlowNet+ [29], DCEIFlow [59], TMA [24], ADM-Flow [31]); Semi-Supervised Learning (SSL), leveraging grayscale images as supervision through photometric loss construction (e.g., EV-FlowNet [18], STE-FlowNet [33]); Unsupervised Learning (USL), relying solely on event data by warping events to reference time and maximizing accumulated image contrast (e.g., MotionPriorCMax [36], ConvGRU-EV-FlowNet [32], FireFlowNet [27], USL-EV-FlowNet [19], ET-FlowNeT [35], EV-MGRFlowNet [63], Paredes *et al.* [37]); and Model-Based (MB) methods, which also adopt contrast maximization objectives but employ traditional nonlinear optimization instead of neural networks (e.g., MultiCM-V2[16], MultiCM [17], Akolkar *et al.* [13], Nagata *et al.* [60], Brebion *et al.* [61], Cuadrado *et al.* [28], Shiba *et al.* [62]). From these paradigms, we select representative prior works as strong baselines for comprehensive quantitative benchmarking of optical flow estimation performance.

Table 1: **Quantitative comparison of optical flow estimation task on MVSEC dataset.** Bold is the best among all methods; underlined is second best. Pink represents the best in the 'USL'; Orange represents the second best in the 'USL'.

| | | indoor_flying1 | | indoor_flying2 | | indoor_flying3 | | outdoor_day1 | | average | |
|---|---|---|---|---|---|---|---|---|---|---|---|
| | $dt=1$ | EPE ↓ | %Out ↓ | EPE ↓ | %Out ↓ | EPE ↓ | %Out ↓ | EPE ↓ | %Out ↓ | EPE ↓ | %Out ↓ |
| SL | EV-FlowNet-EST [25] | 0.97 | 0.91 | 1.38 | 8.20 | 1.43 | 6.47 | – | – | – | – |
| | EV-FlowNet+ [29] | 0.56 | 1.00 | 0.66 | 1.00 | 0.59 | 1.00 | 0.68 | 0.99 | 0.623 | 0.998 |
| | E-RAFT [26] | 1.10 | 5.72 | 1.94 | 30.79 | 1.66 | 25.20 | 0.24 | 1.70 | 1.235 | 15.853 |
| | DCEIFlow [59] | 0.75 | 1.55 | 0.90 | 2.10 | 0.80 | 1.77 | **0.22** | **0.00** | 0.668 | 1.355 |
| | TMA [24] | 1.06 | 3.63 | 1.81 | 27.29 | 1.58 | 23.26 | 0.25 | 0.07 | 1.175 | 13.563 |
| | ADM-Flow [31] | 0.52 | 0.14 | 0.68 | 1.18 | 0.52 | 0.04 | 0.41 | **0.00** | 0.533 | 0.340 |
| SSL | EV-FlowNet [18] | 1.03 | 2.20 | 1.72 | 15.10 | 1.53 | 11.90 | 0.49 | 0.20 | 1.193 | 7.350 |
| | STE-FlowNet [33] | 0.57 | 0.10 | 0.79 | 1.60 | 0.72 | 1.30 | 0.42 | **0.00** | 0.625 | 0.750 |
| MB | Akolkar et al. [13] | 1.52 | – | 1.59 | – | 1.89 | – | 2.75 | – | 1.938 | – |
| | Nagata et al. [60] | 0.62 | – | 0.93 | – | 0.84 | – | 0.77 | – | 0.790 | – |
| | Brebion et al. [61] | 0.52 | 0.10 | 0.98 | 5.50 | 0.71 | 2.10 | 0.53 | 0.20 | 0.685 | 1.975 |
| | Cuadrado et al. [28] | 0.58 | – | 0.72 | – | 0.67 | – | 0.85 | – | 0.705 | – |
| | Shiba et al. [62] | 1.05 | 2.90 | 1.68 | 13.44 | 1.43 | 8.97 | 0.94 | 3.08 | 1.275 | 7.098 |
| | MultiCM [17] | 0.42 | 0.09 | 0.60 | 0.59 | 0.50 | 0.29 | 0.30 | 0.11 | 0.455 | 0.270 |
| | MultiCM-V2 [16] | **0.30** | **0.00** | **0.47** | **0.01** | **0.34** | **0.00** | 0.28 | 0.21 | **0.348** | **0.055** |
| USL | USL-EV-FlowNet [19] | 0.58 | **0.00** | 1.02 | 4.00 | 0.87 | 3.00 | 0.32 | 0.00 | 0.698 | 1.750 |
| | FireFlowNet [27] | 0.97 | 2.60 | 1.67 | 15.30 | 1.43 | 11.00 | 1.06 | 6.60 | 1.283 | 8.875 |
| | ConvGRU-EV-FlowNet [32] | 0.60 | 0.51 | 1.17 | 8.06 | 0.93 | 5.64 | 0.47 | 0.25 | 0.793 | 3.615 |
| | ET-FlowNeT [35] | 0.57 | 0.53 | 1.20 | 8.48 | 0.95 | 5.73 | 0.39 | 0.12 | 0.778 | 3.715 |
| | EV-MGRFlowNet [63] | 0.41 | 0.17 | 0.70 | 2.35 | 0.59 | 1.29 | 0.28 | 0.02 | 0.495 | 0.958 |
| | Paredes et al. [37] | 0.44 | **0.00** | 0.88 | 4.51 | 0.70 | 2.41 | 0.27 | 0.05 | 0.573 | 1.743 |
| | MotionPriorCMax [36] | 0.45 | 0.09 | 0.71 | 2.40 | 0.60 | 0.93 | – | – | – | – |
| | Ours | 0.40 | 0.30 | 0.52 | 0.18 | 0.46 | 0.29 | 0.42 | 0.54 | 0.450 | 0.328 |
| | $dt=4$ | | | | | | | | | | |
| SL | E-RAFT [26] | 2.81 | 40.25 | 5.09 | 64.19 | 4.46 | 57.11 | 0.72 | 1.12 | 3.270 | 40.668 |
| | DCEIFlow [59] | 2.08 | 21.47 | 3.48 | 42.05 | 2.51 | 29.73 | 0.89 | 3.19 | 2.240 | 24.110 |
| | TMA [24] | 2.43 | 29.91 | 4.32 | 52.74 | 3.60 | 42.02 | **0.70** | **1.08** | 2.762 | 31.438 |
| | ADM-Flow [31] | 1.42 | 7.78 | 1.88 | 16.70 | 1.61 | 11.40 | 1.51 | 10.20 | 1.605 | 11.520 |
| SSL | EV-FlowNet [18] | 2.25 | 24.70 | 4.05 | 45.30 | 3.45 | 39.70 | 1.23 | 7.30 | 2.745 | 29.250 |
| | STE-FlowNet [33] | 1.77 | 14.70 | 2.52 | 26.10 | 2.23 | 22.10 | 0.99 | 3.90 | 1.878 | 16.700 |
| MB | Shiba et al. [62] | 4.06 | 53.88 | 6.39 | 71.82 | 5.36 | 65.57 | 3.60 | 49.04 | 4.853 | 60.077 |
| | MultiCM [17] | 1.68 | 12.79 | 2.49 | 26.31 | 2.06 | 18.93 | 1.25 | 9.19 | 1.870 | 16.805 |
| | MultiCM-V2 [16] | **1.18** | **4.77** | **1.87** | **15.51** | **1.38** | **7.62** | 1.05 | 5.68 | **1.108** | **8.305** |
| USL | USL-EV-FlowNet [19] | 2.18 | 24.20 | 3.85 | 46.80 | 3.18 | 47.80 | 1.30 | 9.70 | 2.628 | 32.125 |
| | ConvGRU-EV-FlowNet [32] | 2.16 | 21.51 | 3.90 | 40.72 | 3.00 | 29.60 | 1.69 | 12.50 | 2.688 | 26.083 |
| | ET-FlowNeT [35] | 2.08 | 20.02 | 3.99 | 41.33 | 3.13 | 31.70 | 1.47 | 9.17 | 2.667 | 25.555 |
| | EV-MGRFlowNet [63] | 1.50 | 8.67 | 2.39 | 23.70 | 2.06 | 18.00 | 1.10 | 6.22 | 1.763 | 14.147 |
| | Ours | 1.58 | 9.2 | 2.04 | 18.54 | 1.84 | 13.57 | 1.63 | 14.42 | 1.773 | 13.933 |

For 6-DoF motion estimation, existing methods can be categorized into two classes based on whether depth prior knowledge is required. Approaches such as AEmin [42], Incmin [66], and PEME [43] require depth-augmented event data to achieve a 6-DoF estimation, while ECN [48] and MultiCM-V2 [16] can perform a 6-DoF estimation relying solely on raw event streams without depth priors.

**Implementation Details.** The neural implicit flow field adopts the MLP architecture. The detailed network architecture can be found in appendix. Following prior works [17] [43], we partition the entire event sequence into multiple segments during training, with each segment containing 30k events for MVSEC [64] and 300k events for DSEC [65]. Because the time interval of each segment is short, the cubic spline modeling continuous camera motion employs only 4 control knots, whose 6-dimensional vectors are initialized to a constant value of 0.2. To solve the event warping trajectory Eq. (2), we employ the euler solver for its computational efficiency in numerical integration. The weight of the differential geometric loss is set to 0.25 and the differential flow loss to 1. We employ two separate AdamW optimizers [67] for the neural implicit flow field $NN_\theta$ and the camera motion parameters $\beta$. For the MVSEC dataset, the learning rate for the flow field is exponentially decayed from $1 \times 10^{-4}$ to $6.3 \times 10^{-5}$, whereas for the DSEC dataset, it is cosine annealed from $2 \times 10^{-3}$

Table 2: **Quantitative comparison of optical flow estimation task on DSEC dataset.** Bold is the best among all methods; underlined is second best. Pink represents the best in the 'USL'; Orange represents the second best in the 'USL'.

| Method | All EPE↓ | AE↓ | %Out↓ | interlaken_00_b EPE↓ | AE↓ | %Out↓ | interlaken_01_a EPE↓ | AE↓ | %Out↓ | thun_01_a EPE↓ | AE↓ | %Out↓ |
|---|---|---|---|---|---|---|---|---|---|---|---|---|
| (SL) E-RAFT [26] | 0.79 | 2.85 | 2.68 | **1.39** | 2.36 | 6.19 | 0.90 | 2.54 | 3.91 | 0.65 | 2.94 | 1.87 |
| (SL) TMA [24] | **0.74** | **2.68** | **2.30** | **1.39** | 2.16 | 5.79 | **0.81** | **2.23** | **3.11** | **0.62** | 2.88 | **1.61** |
| (MB) MultiCM-V2 [16] | 3.47 | 13.98 | 30.86 | 5.74 | 9.19 | 38.93 | 3.74 | 9.77 | 31.37 | 2.12 | 11.06 | 17.68 |
| (USL) Paredes et al. [37] | 2.33 | 10.56 | 17.77 | 3.34 | 6.22 | 25.72 | 2.49 | 6.88 | 19.15 | 1.73 | 9.75 | 10.39 |
| (USL) MotionPriorCMax [36] | 3.20 | 8.53 | 15.21 | 3.21 | 4.89 | 20.45 | 2.38 | 5.46 | 17.40 | 1.39 | 6.99 | 7.36 |
| (USL) EV-FlowNet [19] | 3.86 | – | 31.45 | 6.32 | – | 47.95 | 4.91 | – | 36.07 | 2.33 | – | 20.92 |
| (USL) Ours | 3.14 | 10.87 | 19.43 | 7.24 | 14.43 | 35.53 | 3.18 | 7.52 | 19.21 | 1.83 | 6.89 | 12.65 |

| Method | thun_01_b EPE↓ | AE↓ | %Out↓ | zurich_city_12_a EPE↓ | AE↓ | %Out↓ | zurich_city_14_c EPE↓ | AE↓ | %Out↓ | zurich_city_15_a EPE↓ | AE↓ | %Out↓ |
|---|---|---|---|---|---|---|---|---|---|---|---|---|
| (SL) E-RAFT [26] | 0.58 | 2.20 | 1.52 | 0.61 | 4.50 | 1.06 | 0.71 | 3.43 | **1.91** | 0.59 | 2.55 | 1.30 |
| (SL) TMA [24] | **0.55** | **2.10** | **1.31** | **0.57** | 4.38 | **0.87** | **0.66** | 3.09 | 1.99 | **0.55** | 2.51 | **1.08** |
| (MB) MultiCM-V2 [17] | 2.48 | 12.05 | 23.56 | 3.86 | 28.61 | 43.96 | 2.72 | 12.62 | 30.53 | 2.35 | 11.82 | 20.99 |
| (USL) Paredes et al. [37] | 1.66 | 8.41 | 9.34 | 2.72 | 23.16 | 26.65 | 2.64 | 10.23 | 23.01 | 1.69 | 8.88 | 9.98 |
| (USL) MotionPriorCMax [36] | 1.54 | 6.55 | 9.69 | 8.33 | 20.16 | 22.39 | 1.78 | 8.79 | 12.99 | 1.45 | 6.27 | 8.34 |
| (USL) EV-FlowNet [19] | 3.04 | – | 25.41 | 2.62 | – | 25.80 | 3.36 | – | 36.34 | 2.97 | – | 25.53 |
| (USL) Ours | 1.66 | 5.62 | 10.55 | 3.52 | 24.92 | 28.51 | 1.89 | 6.13 | 15.44 | 1.51 | 5.66 | 9.08 |

Table 3: **Quantitative comparison of 6-DoF egomotion estimation task on MVSEC dataset.** The notation "w/ $D$" denotes requiring depth prior information, whereas "w/o $D$" indicates no need for depth information. Bold is the best among all methods; underlined is second best.

| | | indoor_flying1 $RMS_\omega\downarrow$ | $RMS_v\downarrow$ | indoor_flying2 $RMS_\omega\downarrow$ | $RMS_v\downarrow$ | indoor_flying3 $RMS_\omega\downarrow$ | $RMS_v\downarrow$ | outdoor_day1 $RMS_\omega\downarrow$ | $RMS_v\downarrow$ | average $RMS_\omega\downarrow$ | $RMS_v\downarrow$ |
|---|---|---|---|---|---|---|---|---|---|---|---|
| w/ $D$ | AEmin [42] | 1.38 | 0.069 | – | – | – | – | 4.00 | 0.677 | – | – |
| | IncEmin [66] | 1.65 | 0.085 | – | – | – | – | 4.05 | 1.035 | – | – |
| | PEME [43] | **1.05** | **0.039** | – | – | – | – | 2.72 | 0.598 | – | – |
| w/o $D$ | ECN [48] | – | – | – | – | – | – | – | 0.70 | – | – |
| | MultiCM-V2 [16] | 7.72 | 0.24 | 11.50 | 0.27 | 9.53 | 0.31 | 6.85 | 5.90 | 8.900 | 1.680 |
| | Ours | 3.44 | 0.11 | **5.31** | **0.12** | **4.12** | **0.15** | 3.38 | 0.76 | **4.062** | **0.285** |

to $1 \times 10^{-7}$. The learning rate for the camera motion is kept constant at $1 \times 10^{-3}$ for both datasets. Each short event segment is trained for 1k iterations. All experiments were conducted on a NVIDIA RTX 4090.

## 4.2 Results on Optical Flow Estimation

**Quantitative Results.** We conduct comprehensive quantitative benchmarking for optical flow estimation on the MVSEC dataset [64] and DSEC dataset [68], as detailed in Tab. 1 and Tab. 2. The MVSEC benchmarks optical flow estimation over one-frame ($dt = 1$) and four-frame ($dt = 4$) intervals, respectively. Our method outperforms all existing unsupervised learning methods on the MVSEC dataset. Notably, under the $dt = 1$ setting, our method ranks second among all methods, only behind the traditional optimization-based method MultiCM-V2 [16], and outperforms all supervised learning methods. On the DSEC benchmark, our method performs comparably to existing unsupervised learning approaches. However, there remains a performance gap compared to supervised methods. This is primarily because driving scenes involve more complex motion patterns, abrupt illumination changes, and unstructured scene depth, all of which make it more challenging for unsupervised methods to learn accurate optical flow. The experimental results demonstrate that by introducing implicit spatial-temporal continuity constraints Eq. (2) and differential geometric constraints Eq. (9), our method can unlock the network's representation capacity in an unsupervised learning manner, thereby predicting more accurate optical flow.

**Qualitative Results.** We performed a qualitative comparison with strong baselines on the MVSEC dataset and DSEC datasets. As shown in Fig.5, the results on the MVSEC datsets demonstrate that our method predicts optical flow closer to the ground truth. Fig. 6 illustrates that our method also

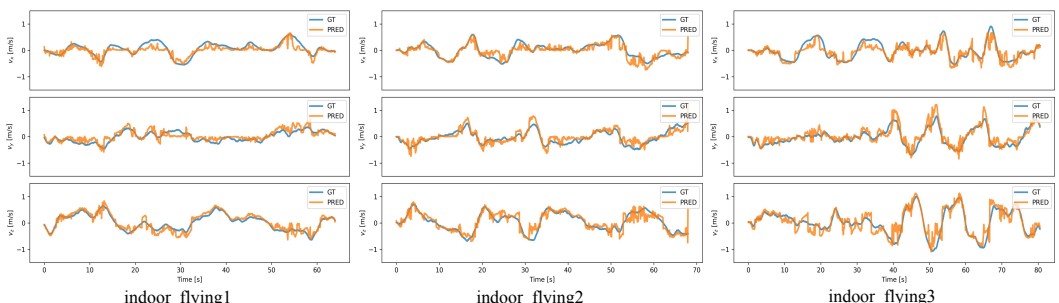

Figure 4: **Qualitative results of 6-DoF motion estimation on the MVSEC dataset.** Due to space constraints, this figure only presents the linear velocity estimation results. The angular velocity results are provided in the supplementary material. The top, middle, and bottom rows in each subfigure correspond to the $x$-axis, $y$-axis, and $z$-axis results, respectively.

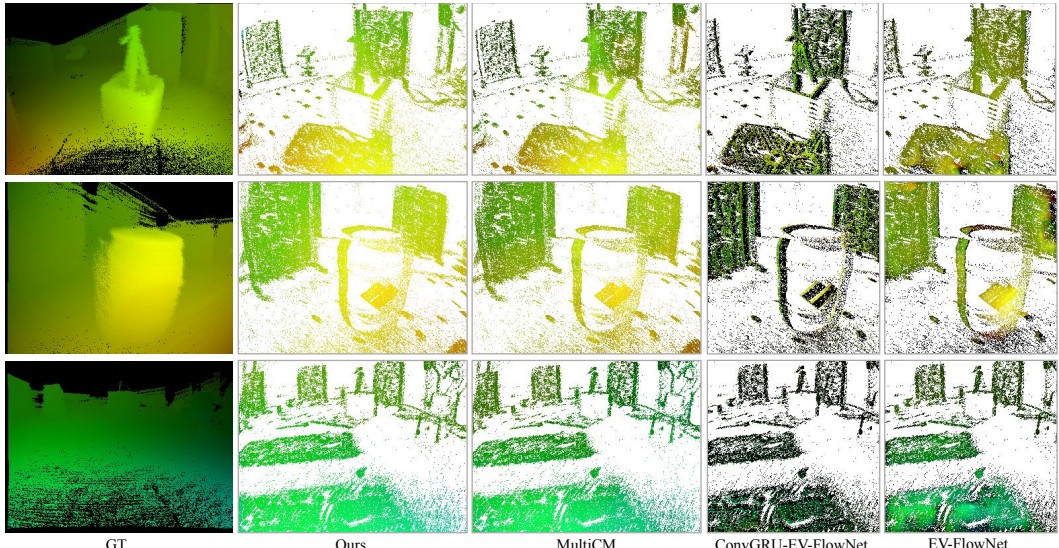

Figure 5: **Qualitative results of optical flow estimation on the MVSEC dataset.** The results in the first, second, and third rows correspond to sequences *indoor_flying1*, *indoor_flying2* and *indoor_flying3*, respectively.

achieved good performance on the DSEC dataset and features smooth optical flow visualizations. Note that we visualize optical flow only at pixels where events are triggered.

### 4.3   Results on 6-DoF EgoMotion Estimation

**Quantitative Results.**   To evaluate the performance of 6-DoF motion estimation, we compared our method with existing approaches on the MVSEC dataset [64], as it provides ground truth camera motion including angular and linear velocities. As shown in Tab. 3, our method achieves state-of-the-art 6-DoF motion estimation performance in both indoor and outdoor scenes. This is primarily attributed to our method's use of splines to represent camera motion Eq. (6), which provides a strong continuity prior, combined with our designed differential geometric loss Eq. (9). This loss enables optical flow information to offer geometrically meaningful guidance for motion learning without requiring depth priors.

**Qualitative Results.**   We also conducted qualitative evaluation on the 6-DoF motion estimation. As illustrated in Fig. 4, the 6-DoF motion estimated by our method closely matches the ground truth, demonstrating that our approach can achieve outstanding motion estimation performance without requiring depth information in an unsupervised paradigm.

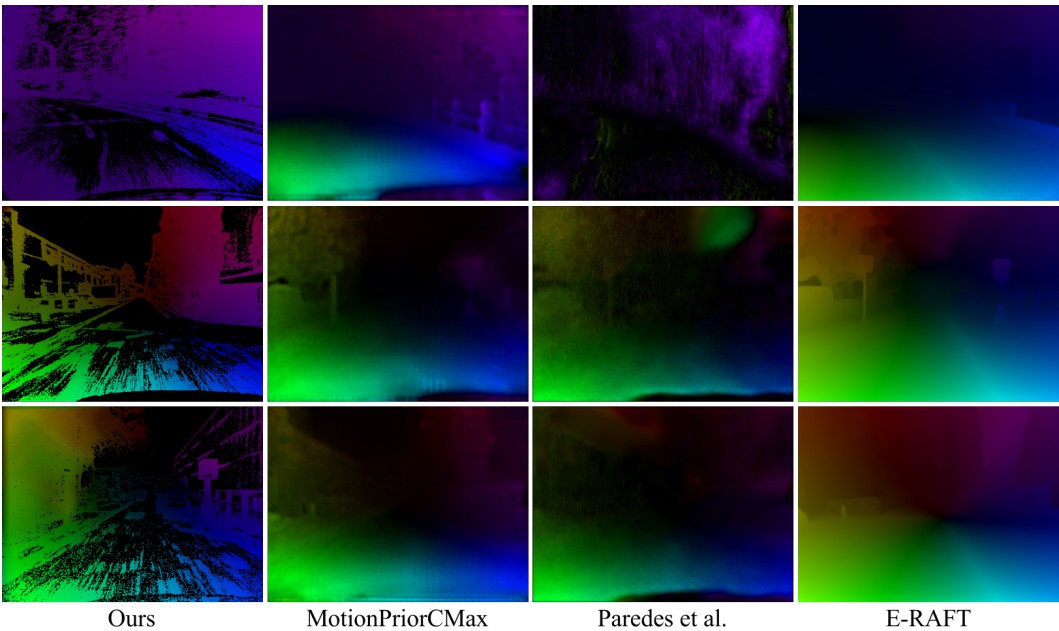

| Ours | MotionPriorCMax | Paredes et al. | E-RAFT |

Figure 6: **Qualitative results of optical flow estimation on the DSEC dataset.** The results in the first, second, and third rows correspond to sequences *zurich_city_15a*, *zurich_city_14c* and *interlaken_01a*, respectively.

Table 4: **Ablation studies on differential geometric constraints.**

| $dt = 4$ | indoor_flying1 | | indoor_flying2 | | indoor_flying3 | | outdoor_day1 | |
| --- | --- | --- | --- | --- | --- | --- | --- | --- |
| | EPE $\downarrow$ | %Out $\downarrow$ | EPE $\downarrow$ | %Out $\downarrow$ | EPE $\downarrow$ | %Out $\downarrow$ | EPE $\downarrow$ | %Out $\downarrow$ |
| w/o geometric constrain | 1.84 | 9.68 | 2.27 | 18.76 | 2.27 | 13.72 | 1.67 | 15.13 |
| w/ geometric constrain | 1.58 | 9.2 | 2.04 | 18.54 | 1.84 | 13.57 | 1.63 | 14.42 |
| ground truth motion | 1.56 | 9.13 | 2.04 | 18.43 | 1.84 | 13.57 | 1.61 | 14.25 |

## 4.4 Ablation Study

To validate the efficacy of differential geometric constraints Eq. (9), we conducted ablation studies on the MVSEC dataset [64] under $dt = 4$ with three configurations: *a)* no geometric constraints, *b)* with differential geometric constraints, and *c)* direct supervised by ground truth motion. As demonstrated in Tab. 4, the differential geometric constraints yield improvements in optical flow estimation, even achieving performance comparable to those supervised with ground truth motion on *indoor_flying2* and *indoor_flying3*. This indicates that our design enhances the geometric plausibility of the estimated optical flow while effectively avoiding convergence to local minima, as visualized in Fig. 3.

## 5 Conclusion

This work presents **E-MoFlow**, a novel framework that unifies 6-DoF egomotion and optical flow estimation using implicit spatial-temporal and geometric regularization within an unsupervised learning paradigm. By incorporating implicit neural representations with differential geometry constraints, our approach effectively tackles the ill-posed challenges of separate estimations of flow and egomotion from event data. Extensive experiments demonstrate that **E-MoFlow** achieves state-of-the-art performance across diverse motion scenarios, matching or surpassing many supervised approaches.

**Acknowledgements.** This work was supported in part by NSFC under Grant 62202389, in part by a grant from the Westlake University-Muyuan Joint Research Institute, and in part by the Westlake Education Foundation.

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

# Appendix

## A Network Architecture

Our network adopts a simple MLP architecture that takes spatial-temporal coordinates $(\mathbf{x}, t)$ as input and outputs optical flow signal $\mathbf{u} = (u, v)$. Compared to [19, 27, 32, 36, 37, 63], this coordinate-based MLP implicitly represents optical flow at spatial-temporal coordinates, essentially a velocity field, without relying on explicit discrete structures (e.g., voxel grid, event count image), enabling temporally continuous and dense flow estimation.

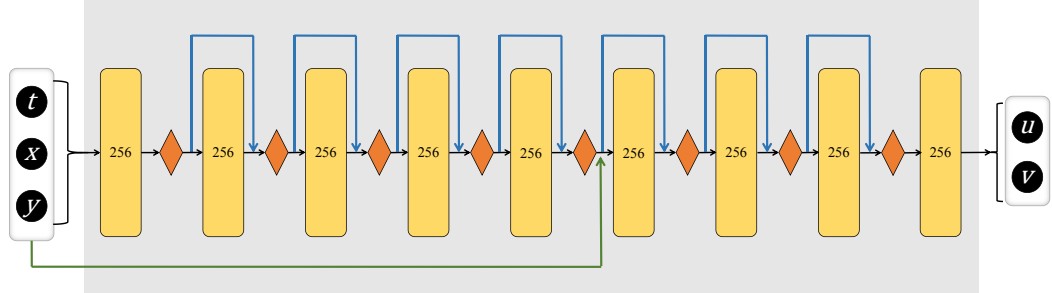

Figure 7: **Schematic Diagram of the Neural Implicit Optical Flow Field Network Architecture.** The input of the network is three-channel spatial-temporal coordinates $(\mathbf{x}, t)$, and the output is optical flow $\mathbf{u} = (u, v)$. The yellow rectangles represent 256-dimensional hidden units. The orange diamonds denote ReLU activation functions. The blue arrows indicate residual connections. The green arrows represent concatenating the original input to the output of the fifth layer.

Specifically, our network architecture, inspired by NeRF [69], employs 9 fully-connected layers with 256 dimensional hidden units. The first eight layers utilize ReLU activations to enforce a low Lipschitz constant, ensuring smoother responses to input variations [70], [71]. This design suppresses high-frequency features while favoring learning of low-frequency features, aligning with the prior that optical flow exhibits spatial-temporal smoothness [17–19]. Notably, no activation function (e.g., ReLU or sigmoid) is applied to the output layer, as optical flow inherently spans both positive and negative values. To further stabilize network training, we introduce residual connections between the second layer to the eighth layer and implement skip connections that concatenate the raw input with the activation outputs of fifth layer. The complete architecture is illustrated in 7.

Although the original NeRF architecture employs positional encoding that enhances high-frequency feature learning [69], our framework deliberately omits such encoding. This design aligns with our goal to model optical flow field which is inherently low-frequency spatial-temporal signals, while avoiding spectral bias toward high-frequency feature [72].

## B Continuous Motion Representation

In this section, we discuss how to select an appropriate motion parameterization $\mathcal{F}$ based on the characteristics of camera egomotion. Given a time $t$, $\mathcal{F}$ maps it to the camera's angular velocity $\boldsymbol{\omega}$ and linear velocity $\boldsymbol{\nu}$ at that moment.

$$\mathcal{F}: t \rightarrow (\boldsymbol{\omega}, \boldsymbol{\nu}), \quad \mathbb{R} \rightarrow \mathbb{R}^3 \times \mathbb{R}^3 \tag{11}$$

In scenarios such as drones, handheld devices, and vehicle-mounted systems, camera ego-motion is constrained by strong prior assumptions. Specifically, camera motion exhibits temporal continuity and smoothness, meaning no abrupt changes occur within infinitesimal time intervals $\Delta t$. This prior is formalized as:

$$\frac{d^k \mathcal{F}}{dt^k} \leq O_k, \quad k \in \{0, 1, 2, \dots, K\} \tag{12}$$

$k$ denotes the order of the derivative and $O$ specifies the upper bounds for their respective derivatives. The equation indicates that the $k$-th order motion derivatives exist and are continuous. This can be

simplified as:

$$\mathcal{F} \in \mathcal{C}^k \tag{13}$$

$\mathcal{C}^k$ denotes the set of functions that have continuous derivatives up to the $k$-th order. Additionally, the motion of the camera is low-dimensional [2]. Thus, there is no need to over-parameterize the camera motion (e.g., using neural networks).

In summary, we employ cubic B-spline as $\mathcal{F}$ to parameterize the camera motion, as its basis functions exhibit $\mathcal{C}^2$ continuity and compact representation via sparse control knots [20]. Specifically, we use four control knots $\beta = [\beta_0, \beta_1, \beta_2, \beta_3]^T \in \mathbb{R}^{4 \times 6}$ over a time interval $t \in [0, 1]$. Therefore, the motion parameterization $\mathcal{F}$ can be formally defined as:

$$
\begin{aligned}
\mathcal{F}(t) &= (\boldsymbol{\omega}_\beta(t), \boldsymbol{\nu}_\beta(t)) \in \mathbb{R}^3 \times \mathbb{R}^3 \\
\boldsymbol{\omega}_\beta(t) &= [\, \mathbf{B}(t)\, \beta \,]_{0:2} \\
\boldsymbol{\nu}_\beta(t) &= [\, \mathbf{B}(t)\, \beta \,]_{3:5}
\end{aligned}
\tag{14}
$$

This definition allows us to derive the camera's angular velocity $\boldsymbol{\omega}_\beta(t)$ and linear velocity $\boldsymbol{\nu}_\beta(t)$ at time $t$, where $\mathbf{B}(t) \in \mathbb{R}^{1 \times 4}$ denotes the cubic B-spline basis functions, defined as follows:

$$
\mathbf{B}(t) = \frac{1}{6} \begin{bmatrix} t^3 & t^2 & t & 1 \end{bmatrix} \begin{bmatrix} -1 & 3 & -3 & 1 \\ 3 & -6 & 3 & 0 \\ -3 & 0 & 3 & 0 \\ 1 & 4 & 1 & 0 \end{bmatrix}
\tag{15}
$$

This design choice inherently satisfies the prior assumptions: 1) Cubic B-spline intrinsically enforces $\mathcal{C}^2$ smoothness priors, ensuring natural continuity in velocity, acceleration and jerk without requiring explicit smoothness constraints. 2) By utilizing sparse control knots, this approach model continuous camera motion while maintaining a low-dimensional parameterization of the 6-DoF egomotion.

## C  Differential Geometric Loss

In 3D vision, the motion of the camera $(\boldsymbol{\omega}, \boldsymbol{\nu})$ induces a motion field $\mathbf{m}$ of projected points on the normalized image plane $\mathbf{x}$. Assuming the camera is a rigid body, the relationship between the motion field and the camera motion can be expressed by the following equation, which we formulate in homogeneous coordinates:

$$\mathbf{m}(\mathbf{x}) = \frac{1}{Z(\mathbf{x})} A(\mathbf{x})\boldsymbol{\nu} + B(\mathbf{x})\boldsymbol{\omega}, \quad \mathbf{x} = [x, y, 1]^T \tag{16}$$

The matrices $A(\mathbf{x})$ and $B(\mathbf{x})$ are functions of homogeneous image coordinates defined as follows:

$$
\boldsymbol{A}(\mathbf{x}) = \begin{bmatrix} -1 & 0 & x \\ 0 & -1 & y \\ 0 & 0 & 0 \end{bmatrix}, \quad \boldsymbol{B}(\mathbf{x}) = \begin{bmatrix} xy & -(1+x^2) & y \\ (1+y^2) & -xy & -x \\ 0 & 0 & 0 \end{bmatrix}
\tag{17}
$$

In practice, $\mathbf{m}(\mathbf{x})$ is approximated by optical flow field $\mathbf{u}(\mathbf{x}) = [u, v, 0]^T$ under brightness constancy assumption.

$$\mathbf{u}(\mathbf{x}) = \frac{1}{Z(\mathbf{x})} A(\mathbf{x})\boldsymbol{\nu} + B(\mathbf{x})\boldsymbol{\omega} \tag{18}$$

Eq.(18) is a critically important motion field equation, which establishes the relationship between optical flow and camera egomotion [22], [73].

However, the presence of $Z(\mathbf{x})$ in this equation implies that recovering camera motion from optical flow or deriving optical flow from camera motion requires knowledge of depth values at each image coordinate. Prior works such as [14, 15, 40–42], rely on depth priors or assume locally shared depth values when performing 6-DOF motion estimation, while methods like [16, 19, 48] jointly estimate depth alongside optical flow and 6-DOF motion. However, this expands the parameterization space of the optimization problem, introducing additional degrees of freedom that may lead to convergence to local minima. Therefore, to enable the formulation of an unsupervised loss function that can simultaneously estimate optical flow and 6-DOF motion with high accuracy, we need to eliminate the dependence on $Z(\mathbf{x})$.

Table 5: **Ablation studies on early stopping strategy.**

| | | indoor_flying1 | indoor_flying2 | indoor_flying3 | outdoor_day1 |
|---|---|---|---|---|---|
| Time | w/o early stopping | 9.30s | 9.41s | 9.50s | 10.77s |
| | w/ early stopping | 4.21s | 4.30s | 4.87s | 4.56s |
| | efficiency improvement | 2.21× ↑ | 2.19× ↑ | 1.99× ↑ | 2.36× ↑ |
| EPE | w/o early stopping | 1.58 | 2.04 | 1.84 | 1.63 |
| | w/ early stopping | 1.61 | 2.09 | 1.90 | 1.68 |
| | performance drop | 1.90% ↓ | 2.45% ↓ | 3.26% ↓ | 3.07% ↓ |

We transpose Eq.(18) and then left-multiply by $\boldsymbol{\nu} \times \mathbf{x}$, form the inner product of $\mathbf{u}(\mathbf{x})$ and $\boldsymbol{\nu} \times \mathbf{x}$, yieding a scalar equation to isolate $Z(\mathbf{x})$ as follows. $\times$ denotes the cross product operation.

$$\mathbf{u}(\mathbf{x})^T \left(\boldsymbol{\nu} \times \mathbf{x}\right) = \left(\frac{1}{Z(\mathbf{x})} A(\mathbf{x})\boldsymbol{\nu} + B(\mathbf{x})\boldsymbol{\omega}\right)(\boldsymbol{\nu} \times \mathbf{x}) \tag{19}$$

Simplify the above equation to obtain:

$$\mathbf{u}(\mathbf{x})^T[\boldsymbol{\nu}]_\times \mathbf{x} = \frac{1}{Z(\mathbf{x})} \boldsymbol{\nu}^T A(\mathbf{x})^T[\boldsymbol{\nu}]_\times \mathbf{x} + \boldsymbol{\omega}^T B(\mathbf{x})^T[\boldsymbol{\nu}]_\times \mathbf{x} \tag{20}$$

where $[\cdot]_\times$ denotes the skew-symmetric operation. Interestingly, it can be proven that the coefficient of the term that involves $Z(\mathbf{x})$ in Eq.(20) is identically zero.

$$\boldsymbol{\nu}^T A(\mathbf{x})^T[\boldsymbol{\nu}]_\times \mathbf{x} \equiv 0 \tag{21}$$

Therefore, Eq.(20) can be further simplified as follows:

$$\mathbf{u}(\mathbf{x})^T[\boldsymbol{\nu}]_\times \mathbf{x} - \boldsymbol{\omega}^T B(\mathbf{x})^T[\boldsymbol{\nu}]_\times \mathbf{x} = 0 \tag{22}$$

By expanding $\boldsymbol{\omega}^T B(\mathbf{x})^T[\boldsymbol{\nu}]_\times \mathbf{x}$, Eq.(22) can be rewritten in the following form:

$$\mathbf{u}(\mathbf{x})^T[\boldsymbol{\nu}]_\times \mathbf{x} - \mathbf{x}^T \mathbf{s}\mathbf{x} = 0, \quad \mathbf{s} = \frac{1}{2}\left([\boldsymbol{\omega}]_\times[\boldsymbol{\nu}]_\times + [\boldsymbol{\nu}]_\times[\boldsymbol{\omega}]_\times\right) \tag{23}$$

Finally, we obtained an equation that connects the optical flow field and camera motion without relying on depth values. Eq.(23) can theoretically be regarded as a differential form of the epipolar constraint. We use this as our differential geometric loss to jointly learn optical flow and 6-DoF motion, as shown in the following equation.

$$L_{\text{geometry}}(t, \mathbf{x}, \theta, \beta) = \left\| \mathbf{u}_\theta(t, \mathbf{x})^T[\boldsymbol{\nu}_\beta(t)]_\times \mathbf{x} - \mathbf{x}^T \mathbf{s}_\beta(t)\mathbf{x} \right\|_2^2,$$
$$\mathbf{s}_\beta(t) = \frac{1}{2}\left([\boldsymbol{\omega}_\beta(t)]_\times[\boldsymbol{\nu}_\beta(t)]_\times + [\boldsymbol{\nu}_\beta(t)]_\times[\boldsymbol{\omega}_\beta(t)]_\times\right) \tag{24}$$

Here, $\mathbf{u}_\theta(t, \mathbf{x})$ represents the optical flow obtained from our neural implicit representation, while $\boldsymbol{\omega}_\beta(t)$ and $\boldsymbol{\nu}_\beta(t)$ denote the angular velocity and linear velocity of the camera, derived from the cubic B-spline continuous motion representation.

## D   More Ablation Studies

**Early Stopping Strategy.**   To enhance computational efficiency, we employed the early-stopping strategy from [52–54] on the MVSEC dataset [64] under $dt = 4$. Specifically, we set the patience to 45 and the minimum improvement threshold to $1 \times 10^{-3}$, applying the early stopping strategy after 300 iterations. The results in Tab. 5 indicate that the training speed was boosted by 2.2x, while the optical flow estimation accuracy showed only a slight 2.67% drop in the EPE. This demonstrates that the strategy significantly reduces training time at the cost of only a minor loss in accuracy, achieving an excellent trade-off.

## E   More Qualitative Results

We further provide additional qualitative results. As shown in 8, our method achieves comprehensive 6-DoF motion estimation on the MVSEC dataset [64]. The angular velocity and linear velocity estimated by our approach closely match the ground-truth motion.

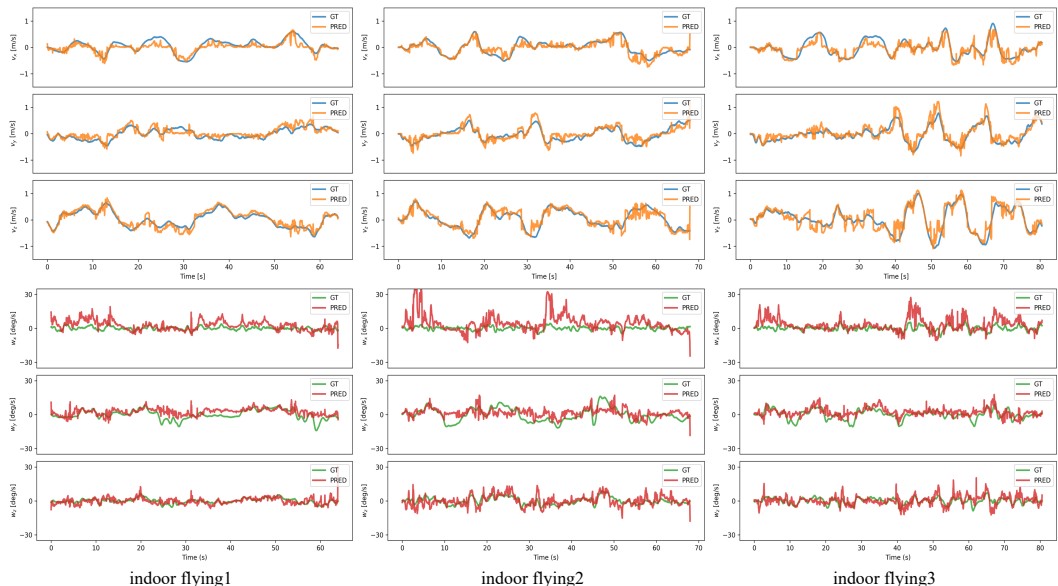

Figure 8: **Complete qualitative results of 6-DoF motion estimation on the MVSEC dataset.** The top section displays the linear velocity estimation results (in $m/s$), while the bottom section shows the angular velocity estimation results (in deg/$s$). The top, middle, and bottom rows in each subfigure correspond to the $x$-axis, $y$-axis, and $z$-axis results, respectively.

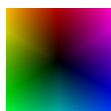

Figure 9: **Color wheel for visualizing optical flow.** A green color in the optical flow visualization corresponds to motion directed toward the lower-left corner of the image, while the saturation of the color encodes the flow magnitude — more vivid hues indicate larger displacement values.

For MVSEC datasets [64], We provide additional qualitative comparisons of optical flow estimation between our method and MultiCM [17], the second-best performing baseline. As shown in 10, our approach predicts optical flow with superior continuity and smoothness, validating the effectiveness of our neural implicit optical flow field representation. The color wheel used to visualize optical flow is shown in 9, where different colors encode the magnitude and direction of the optical flow.

For DSEC datasets [68], We provide additional visualization comparisons of optical flow estimation between our method, state-of-the-art unsupervised learning methods, and supervised learning methods on more sequences. The results in Fig. 11 demonstrate that in some scenarios, our method yields visually superior results with smoother optical flow, while in scenes with complex textures and drastic depth changes, it may produce errors at the edges.

Furthermore, we also provide visualization results of the flow field in $X - Y - T$ 3D space on the MVSEC dataset [64]. The results in Fig. 12 indicate that our method exhibits an emergent capability for point tracking, which is the ability to track the movement of points in pixel space.

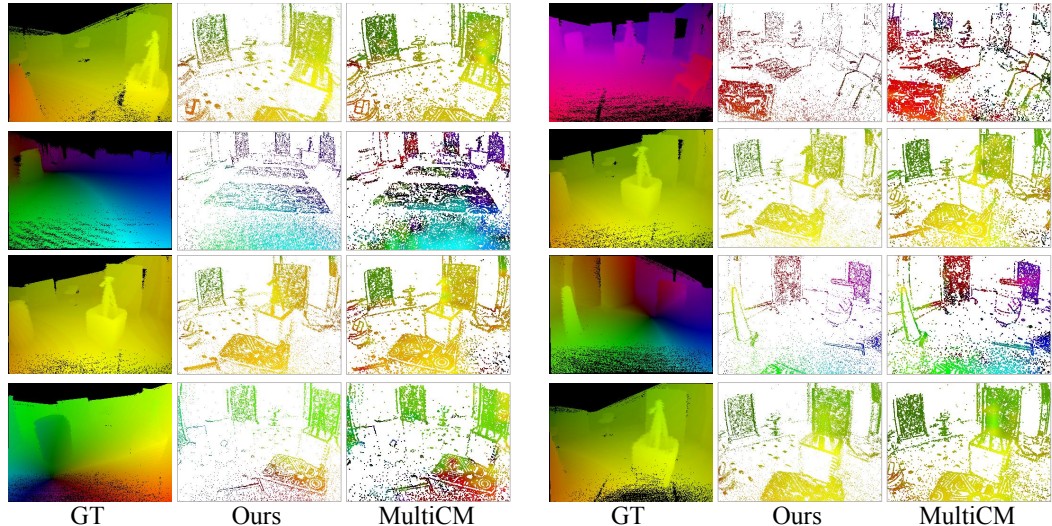

| GT | Ours | MultiCM | GT | Ours | MultiCM |

Figure 10: **More qualitative results of optical flow estimation on the MVSEC dataset.** It can be clearly observed that our method estimates smoother optical flow, free from abrupt variations, and demonstrates closer alignment with the ground truth optical flow. This indicates that our approach more effectively models the intrinsically spatial-temporally continuous optical flow field.

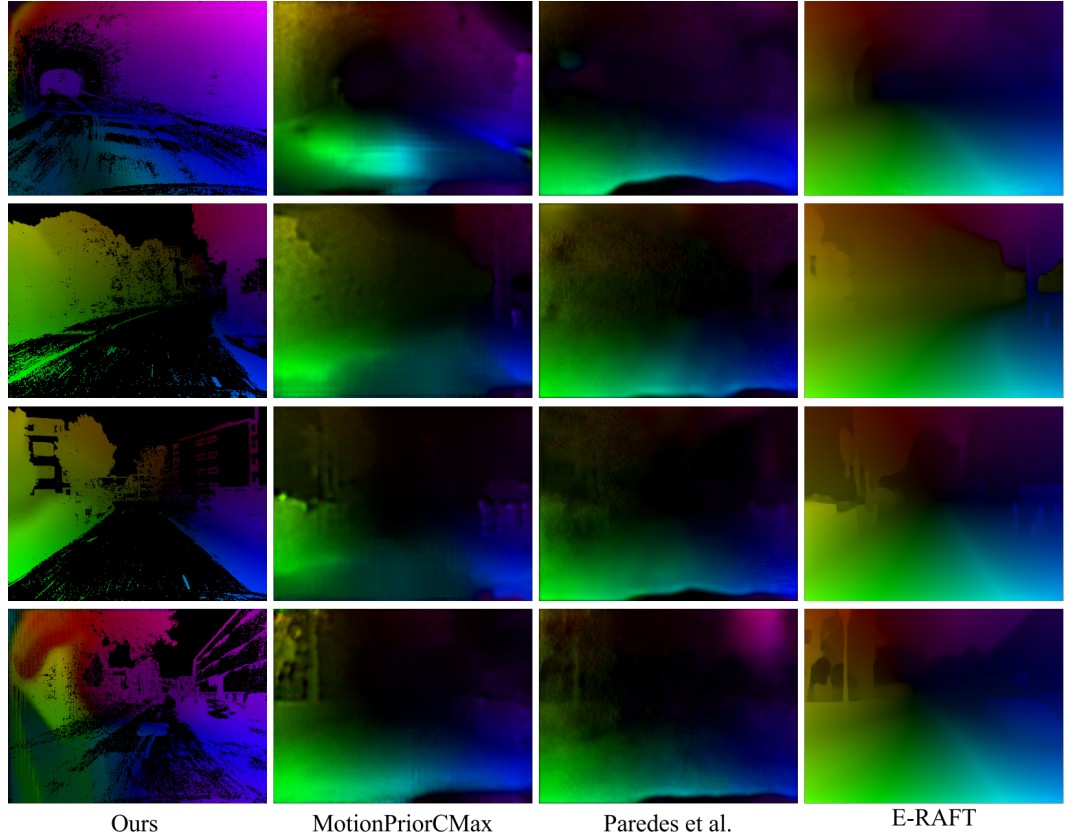

| Ours | MotionPriorCMax | Paredes et al. | E-RAFT |

Figure 11: **More qualitative results of optical flow estimation on the DSEC dataset.** Qualitative results of optical flow estimation on the DSEC dataset. The results in the first, second, third, and fourth rows correspond to sequences *interlaken_00b*, *thun_01a*, *thun_01a*, and *zurich_city_14a* respectively.

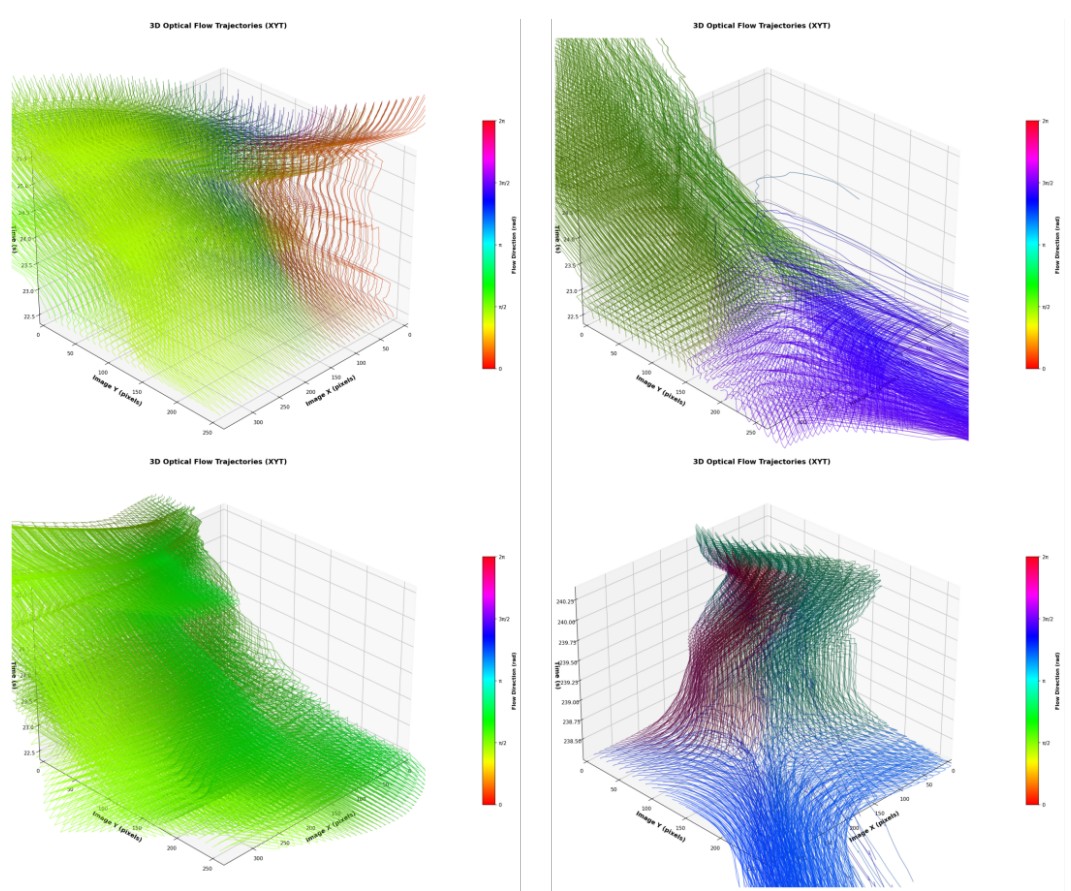

Figure 12: **Point tracking results of flow field on the MVSEC dataset.**

