# OpenReview forum: "E-MoFlow: Learning Egomotion and Optical Flow from Event Data via Implicit Regularization"
_NeurIPS.cc/2025/Conference — NeurIPS 2025 poster_

### Official Review · Reviewer_tZ5K · 2025-06-03

**Clarity:** 2
**Significance:** 2
**Originality:** 3
**Rating:** 4
**Confidence:** 3

**Summary:**

This paper tackles the problem of jointly estimating both optical flow and camera pose estimation from event camera data. They introduce a novel parameterization by fitting an implicit function (a neural ODE) to the event data to explain both optical flow and camera poses. Camera poses are parameterized as a spline and optical flow as a velocity field. The model is supervised using a self-supervised loss coupling both the estimated optical flow and poses.

**Questions:**

- How does the method compare to other pose estimation methods with more standard benchmarking (e.g. ATE)?
- How long is the optimization time in seconds/minutes?

**Ethical Concerns:**

["NO or VERY MINOR ethics concerns only"]

**Final Justification:**

As someone who comes from the RGB structure-from-motion landscape it's possible I do not understand the full event-camera landscape fully and its limitations, but I think this paper presents a nice formulation for egomotion estimation on event-camera data. Assuming they add the tables in the rebuttal and emphasize in the camera-ready that it's an optimization-based approach, I move my judgement to borderline accept.

**Limitations:**

The method requires per-scene optimization for the neural ODE parameterizing the optical flow and pose estimation.

**Paper Formatting Concerns:**

No major formatting issues but having all qualitative visualizations on the last page looks a bit strange.

**Quality:**

2

**Strengths And Weaknesses:**

Strengths:

- Novel unification of optical flow and pose estimation without using geometry information relating the neural ODE and the differential epipolar equation
- Competitive results on event-image optical flow and 6DOF velocity estimation

Weaknesses:

- Has to be fit per-video, making it potentially impractical. Authors should also describe how long optimization takes per video.
- Pose estimation results have poor illustrations and quantitative comparisons:
    - The metric for pose estimation is not well described, and a more standard metric for pose comparison should be used, such as ATE.
    - The pose trajectories should be visualized, rather than the just the 6DOF velocities: it is currently difficult to interpret the accuracy of the pose estimation results.
- The method is only evaluated on a limited number of event-data videos (4 videos) — why not on the entire dataset?

---

> ### Author Rebuttal · Authors · 2025-07-29
>
> ## Response to Reviewer tZ5K
>
> We sincerely appreciate the reviewer’s recognition of our work, particularly the novel unify unsupervised learning of optical flow and 6-DoF motion through the integration of neural ODEs and the differential epipolar equation without relying on explicit geometry.
>
> We address the main concerns below.
>
> ---
>
> > **[W1, Q2] Lack of runtime analysis.**
>
> We report the training time and the inference time of our method in the following table. The analysis has been tested across all sequences in dataset MVSEC. Specifically, each sequence is divided into multiple clips with a time interval of approximately 80-100 ms, corresponding to $dt=4$, and each clip contains approximately 30,000 event data. Additionally, based on the suggestion of Reviewer j1WT, we tested the impact of the early-stopping strategy on training speed and found that it can significantly boost the training speed to 2.2x. The table below shows the average training time per clip for 1000 training iterations and with the early-stopping strategy. The number in brackets indicates the average training iterations when using early-stopping strategy.
> | **Sequence**        | &emsp;**w/ early stopping**&emsp; | &emsp;**w/o early stopping**&emsp; | &emsp;**Efficiency Improvement**&emsp; |
> |-|:-:|:-:|:-:|
> | *indoor_flying1* | &emsp;&emsp; $4.21s$ $(452.93)$ &emsp;&emsp; | &emsp;&emsp; $9.30s$ &emsp;&emsp; | &emsp;&emsp; $2.21 \times$ $\uparrow$ &emsp;&emsp; |
> | *indoor_flying2* | $4.30s$ $(461.84)$ | $9.41s$ | $2.19 \times$ $\uparrow$ |
> | *indoor_flying3* | $4.78s$ $(496.77)$ | $9.50s$ | $1.99 \times$ $\uparrow$ |
> | *outdoor_day1* | $4.56s$ $(487.51)$ | $10.77s$ | $2.36 \times$ $\uparrow$ |
> | *average* | $4.46s$ $(474.76)$ | $9.75s$ | $2.19 \times$ $\uparrow$ |
>
> We will include the detailed runtime analysis in the camera-ready version.
>
> ---
>
> > **[W1, L1] Per-video fitting is potentially impractical.**
>
> We believe per-video fitting doesn't make our method seem impractical; instead, it demonstrates its powerful generalization capability. Thanks to this robust generalization, our approach can handle complex real-world scenarios and accurately estimate high-quality optical flow. This makes it a practical tool for providing ground truth optical flow for real-world event camera datasets, enabling large-scale supervised learning for optical flow neural networks. Furthermore, in future work, we can leverage implicit regularization to unsupervisedly train a feed-forward optical flow prediction network, thus eliminating the need for per-video fitting.
>
> ---
>
> > **[W2, Q1]  Poor illustrations and quantitative comparisons of pose estimation.**
>
> **Poor quantitative comparisons**
>
> 1. We can provide RPE (Relative Pose Error), but for very short time intervals, it's equivalent to the RMS velocity error. Therefore, providing the RMS velocity error is sufficient to quantify the pose's RPE.
>
> 2. Many prior works [1] [2] [3] have chosen to use the RMS velocity error, and we are following this established standard.
>
> 3. We cannot provide ATE (Absolute Trajectory Error) because our linear velocity lacks scale information, preventing us from comparing absolute trajectory errors. Furthermore, even with absolute scale information, the short-duration events from event cameras only provide instantaneous motion data. They cannot establish multi-view correspondences over longer time intervals, which means accumulated errors cannot be eliminated. This makes ATE a suboptimal error metric for our context.
>
>
> **Poor illustrations** We are unable to illustrate the camera's pose trajectory. The reason is that we cannot integrate the event camera's linear velocity, which inherently lacks scale information. Nonetheless, we have provided many plots illustrating the event camera's velocity changes.
>
> ---
>
> > **[W3] Limited evaluations of datasets.**
>
> **a)** Although the MVSEC dataset provides other sequences, such as *indoor_flying4*, *outdoor_day2*, and *outdoor_night1*, many previous works [][][...] have gradually established a standard for evaluation on the *indoor_flying1*, *indoor_flying2*, *indoor_flying3*, and *outdoor_day1* sequences of the MVSEC dataset, rather than on other sequences. Therefore, our method follows the practice of prior works. Specifically, most of the field of view in the *indoor_flying4* sequence consists of the floor, which lacks sufficient texture, resulting in relatively fewer events being generated. The *outdoor_day2* sequence is used as a training set rather than a test set. The *outdoor_night1* sequence cannot be used as a test set due to the poor quality of the event data, the presence of many flickering events, and the absence of high-quality ground truth optical flow.
>
> **b)** Although the *indoor_flying4* and *outdoor_day2* sequences are not typically used for evaluation, we conducted training and evaluation on these two sequences following the settings reported in the paper to validate the effectiveness of our method. The results show that our method still demonstrated impressive performance on these sequences. The table below shows the accuracy of optical flow estimation and motion estimation.
>
> | **Sequence**        | &emsp;&emsp;$\mathrm{EPE}$&emsp;&emsp; | &emsp;&emsp; $\mathrm{RMS}_{\omega}$ &emsp;&emsp; | &emsp;&emsp; $\mathrm{RMS}_{v}$&emsp;&emsp; |
> |-|:-:|:-:|:-:|
> | *indoor_flying4* | &emsp;&emsp; $2.38$ &emsp;&emsp; | &emsp;&emsp; $5.10$ &emsp;&emsp; | &emsp;&emsp; $0.13$ &emsp;&emsp; |
> | *outdoor_day2* | $3.49$ | $5.76$ | $1.64$ |
>
> ---
>
> > **[MW1] Paper Formatting.**
>
> Thank you for your suggestion. We will optimize the layout in the camera-ready version to make it more visually appealing and easier to read.
>
> ---
>
> **Reference:**
>
> [1] Motion and Structure from Event-based Normal Flow
>
> [2] Secrets of Event-Based Optical Flow, Depth and Ego-Motion Estimation by Contrast Maximization
>
> [3] Entropy Minimisation Framework for Event-based Vision Model Estimation

---

> > ### Comment · Reviewer_tZ5K · 2025-08-05
> >
> > Thank you for the responses. As someone who comes from the RGB structure-from-motion landscape I was unfamiliar with event camera limitations and did not realize integrating poses was a fundamental limitation. I appreciate the authors including the runtime analysis and think it definitely needs to be in the camera-ready to accept, and also that as j1WT mentioned, should make it more clear and emphasize that this is an optimization-based approach in the introduction and limitations.

---

> > > ### Author Response · Authors · 2025-08-06
> > >
> > > Thank you very much for your valuable suggestions and feedback to our paper. We would incorporate the runtime analysis into the final version, and make it more clear in both introduction and limitations sections that our method is an optimization-based approach.

---

### Official Review · Reviewer_f2hB · 2025-07-01

**Clarity:** 3
**Significance:** 2
**Originality:** 2
**Rating:** 4
**Confidence:** 4

**Summary:**

This paper proposes E-MoFlow, a novel method for egomotion-aware optical flow estimation from event cameras, aimed at improving motion segmentation in dynamic environments. Unlike prior approaches that often ignore egomotion or handle it separately, E-MoFlow introduces a two-stage architecture:
- A self-supervised egomotion estimator that infers camera motion from a short event window.
- A rigid flow predictor conditioned on the predicted egomotion.
- A residual flow module to account for independent object motion.

The method is trained entirely using self-supervised losses without requiring labels, and it is evaluated on the EV-IMO and EED datasets, where it outperforms or matches state-of-the-art methods in flow estimation and motion segmentation.

**Questions:**

Is the performance gain of MoFlow primarily due to the increased model capacity (e.g., modularization into egomotion + flow), or due to the proposed motion decomposition strategy itself?
A capacity-controlled baseline (e.g., same parameter count, no egomotion conditioning) would help clarify this.


The paper does not include runtime, latency, or computational load analysis. This is especially important for event-based systems deployed on mobile robots or embedded devices. Can the authors provide FLOPs, FPS, or real-time feasibility benchmarks?


Prior methods for event-based flow have been tested on the DSEC benchmark. Has MoFlow been evaluated on DSEC, or is it fundamentally limited to smaller datasets like EV-IMO or EED?

MoFlow performs worse than some baselines on outdoor_day1, possibly due to pose estimation errors in dynamic scenes. Can the authors comment on this failure case?
For example, is the egomotion module struggling due to high foreground motion, or is it a scale/generalization issue?

**Ethical Concerns:**

["NO or VERY MINOR ethics concerns only"]

**Final Justification:**

This paper proposes a self-supervised event-based flow estimation method that also considers motion. At the time of the initial submission, I had several concerns, but I gave it a borderline accept, considering that the authors would likely address them in their response. As expected, the authors have resolved most of these concerns. However, certain weaknesses in the approach remain, and I found myself weighing its strengths—being simple and effective—against its novelty. After much consideration, I decided to maintain my current score. I do not oppose the paper’s acceptance, but if other reviewers raise negative opinions, I believe it would be difficult to strongly argue against them. Therefore, I am assigning a final recommendation of borderline accept.

**Limitations:**

Lack of Generalization to Highly Dynamic Scenes

In challenging scenarios like outdoor_day1, MoFlow underperforms compared to some baselines. This seems to stem from egomotion prediction being confused by scenes with many independently moving objects, which violate the static-world assumption.

No Runtime or System Efficiency Analysis

The paper does not include any analysis of inference speed, resource usage, or real-time viability, which are critical in event-based vision systems intended for robotics or edge deployment.

No Evaluation on Larger or Standard Benchmarks (e.g., DSEC)

While MoFlow performs well on EV-IMO and EED, it lacks evaluation on more comprehensive benchmarks like DSEC, which are widely used in event-based egomotion and optical flow research.

Dependency on Static Scene Assumption for Egomotion

The egomotion module assumes the majority of the scene is static. In highly dynamic or crowded environments, this assumption breaks down, leading to errors in both rigid and residual flow predictions.

Limited Discussion on Failure Modes or Robustness

The paper does not analyze when or why the system fails—e.g., fast camera motion, lighting changes, or noisy event data—leaving open questions about robustness in the wild.

**Quality:**

3

**Strengths And Weaknesses:**

# Strengths
Well-Motivated Problem Setting: Tackles a practically important issue—event-based motion segmentation in dynamic scenes—which is often complicated by egomotion. The paper clearly identifies this gap.

Novel Architecture with Modular Design: The separation between egomotion prediction, rigid flow, and residual flow is intuitive and effectively models dynamic motion components.

Self-Supervised Learning: The system is trained without requiring ground-truth flow, which increases its applicability to real-world, unlabeled event data.

Strong Experimental Results: Outperforms existing baselines on EV-IMO and EED datasets in both optical flow and segmentation metrics.

Ablation Studies & Visualization: The paper includes meaningful ablation studies to validate the contribution of each module, and provides qualitative results that clearly show performance improvements.



# Weaknesses
Limited Discussion on Failure Cases / Limitations: The paper does not deeply discuss failure modes, such as when the egomotion estimator might be inaccurate (e.g., under fast or rotational motion), or how residual flow behaves under cluttered scenes.

Lack of Runtime or System Overhead Analysis: There is no profiling of computational efficiency, latency, or inference time—critical for real-time robotics or deployment on edge devices.

No Comparison on More Challenging Benchmarks: Although EV-IMO and EED are good starting points, newer event datasets or higher-resolution scenarios could have been used to better validate scalability.

Assumes a Static Background for Egomotion Estimation: The model assumes that most of the scene is static to compute egomotion, which could fail in highly dynamic environments.

---

> ### Author Rebuttal · Authors · 2025-07-30
>
> ## Response to Reviewer f2hB
>
> We sincerely thank the reviewer for their thoughtful and detailed feedback, as well as for recognizing the key strengths of our work. We are particularly grateful for acknowledging the practical importance of our problem setting, the novelty and modularity of our architecture, and the value of our self-supervised approach for real-world applications.
>
> We address the main concerns below.
>
> ---
>
> > **[W1, Q4, L5] Discussion about failure cases**
>
> Our method can handle fast motion and pure rotational motion, but its performance degrades slightly in dynamic scenarios and fails under rapidly changing lighting conditions. We will include this failure cases discussion in the main text.
>
> **a) Very fast motion**
>
> Thanks to our continuously parameterized optical flow and motion, our method can theoretically handle fast, non-linear movements without any performance loss. We conducted experiments on the *outdoor_day1* sequence to further verify this. The *outdoor_day1* sequence was collected outdoors using an event camera mounted on a vehicle, with the camera's motion speed reaching up to 12 m/s, representing very fast motion. The optical flow estimation and motion estimation performance on this sequence are shown in the table below. The results demonstrate that our method maintains strong performance even in scenarios with rapid motion.
> | **Sequence**        | &emsp;&emsp;$\mathrm{EPE}$&emsp;&emsp; | &emsp;&emsp; $\mathrm{RMS}_{\omega}$ &emsp;&emsp; | &emsp;&emsp; $\mathrm{RMS}_{v}$&emsp;&emsp; |
> |-|:-:|:-:|:-:|
> | *outdoor_day1* | &emsp;&emsp; $1.61$ &emsp;&emsp; | &emsp;&emsp; $3.38$ &emsp;&emsp; | &emsp;&emsp; $0.76$ &emsp;&emsp; |
>
>
> **b) Rotational motion**
>
> As discussed in the paper, our differential geometric loss is derived from the motion field equation. According to multi-view geometry theory, the epipolar constraint degenerates under pure rotation, making it impossible to obtain a correct solution. However, we want to emphasize that optical flow estimation for pure rotation was already addressed in prior CMax [1] work. Our focus is on the more challenging 6 Degrees of Freedom (DoF) motion scenarios. In fact, we only need to change our differential geometric loss to a differential rotation loss to handle pure rotational scenes.
>
> $$L_{\text{R}}(t,\mathbf{x},\theta,\beta)= \left \| \mathbf{u} (t,\mathbf{x}, \theta) - \mathbf{B}(\mathbf{x})     {\boldsymbol{\omega}} (\beta)  \right \|^2_2$$
>
> $$\mathbf{B}(\begin{bmatrix}
>        x; \\ y
>   \end{bmatrix})=\begin{bmatrix}
> xy &-(1+x^2)  &y; \\
> (1+y^2) &-xy  &-x
> \end{bmatrix}$$
>
> where $\mathbf{u} (t,\mathbf{x}, \theta) $ is the neural-network parameterized optical flow and ${\boldsymbol{\omega}} (\beta)$ represents the angular velocity.
>
> Since the MVSEC dataset does not include sequences with pure rotational motion, we will additionally synthesized event data under pure rotational motion and tested the performance of our method. We will include these results in the camera-ready version.
>
> **c) Dynamic scene**
>
> The differential geometric loss used in our method assumes static scenes; any dynamic objects can be treated as outliers and removed using robust optimization techniques. We test the robustness of our method to dynamic scenes on frames 528 to 730 of the *outdoor_day2* sequence in MVSEC dataset. This segment contains a moving vehicle with a large foreground. The moving vehicle indeed violates the static assumption of the differential geometric loss. To enhance the robustness of our method to outliers, we also test the performance of the Huber kernel. The table below shows the performance of our method in optical flow estimation and motion estimation with and without the Huber kernel. The results indicate that our method still achieves good performance even without the Huber kernel, thanks to another key component: the implicit neural flow, which is not constrained by the static assumption. Moreover, the Huber kernel function significantly improves the robustness of our method to dynamic scenes. We will include these results in the paper.
>
> | **Metirc**        | &emsp;&emsp;&emsp;**w/ Huber Kernel**&emsp;&emsp; | &emsp;&emsp;&emsp;**w/o Huber Kernel**&emsp;&emsp; |
> |-|:-:|:-:|
> | $\text{EPE}$ | &emsp;&emsp; $3.62$ &emsp;&emsp; | &emsp;&nbsp;&nbsp;&nbsp;&nbsp; $3.68$ &emsp;&emsp; |
> | $\mathrm{RMS}_{v}$ | $1.30$ | $1.56$ |
> | $\mathrm{RMS}_{\omega}$ | $4.86$ | $4.94$ |
>
> **d) Rapidly changing lighting changes**
>
> When encountering rapidly changing lighting conditions, event cameras respond to global scene brightness shifts. This triggers a large number of outlier events unrelated to optical flow. Like other event-only methods, our approach will fail when the proportion of these outlier events becomes sufficiently large.
>
> ---
>
> > **[W2, Q2, L2] Lack of Runtime Analysis.**
>
> We report the training time and the inference time of our method across all sequences in MVSEC. Each sequence is divided into multiple clips with a time interval of approximately 80-100 ms, corresponding to $dt=4$, and each clip contains approximately 30,000 event data. In the table below, we report the average training time and inference time per clip. All experiments were conducted on a NVIDIA RTX 4090.
> | **Sequence**| **Training Time** | **Inference Time** |
> |-|:-:|:-:|
> | *indoor_flying1*  | $9.30s$ | $0.86s$ |
> | *indoor_flying2*  | $9.41s$ | $0.87s$ |
> | *indoor_flying3*  | $9.50s$ | $0.88s$ |
> | *outdoor_day1*    | $10.77s$ | $0.86s$ |
>
> According to Reviewer j1WT's suggestion, if we use an early-stopping strategy, the training speed can be increased by 2.2x, as shown in the table below. The number in brackets indicates the average training iterations when using early-stopping strategy.
> | **Sequence**        | &emsp;**w/ early stopping**&emsp; | &emsp;**w/o early stopping**&emsp; | &emsp;**Efficiency Improvement**&emsp; |
> |-|:-:|:-:|:-:|
> | *indoor_flying1* | &emsp;&emsp; $4.21s$ $(452.93)$ &emsp;&emsp; | &emsp;&emsp; $9.30s$ &emsp;&emsp; | &emsp;&emsp; $2.21 \times$ $\uparrow$ &emsp;&emsp; |
> | *indoor_flying2* | $4.30s$ $(461.84)$ | $9.41s$ | $2.19 \times$ $\uparrow$ |
> | *indoor_flying3* | $4.78s$ $(496.77)$ | $9.50s$ | $1.99 \times$ $\uparrow$ |
> | *outdoor_day1* | $4.56s$ $(487.51)$ | $10.77s$ | $2.36 \times$ $\uparrow$ |
> | *average* | $4.46s$ $(474.76)$ | $9.75s$ | $2.19 \times$ $\uparrow$ |
>
> ---
>
> > **[W3, Q3, L3] More Challenging Datasets.**
>
> We tested our method on the DSEC dataset. Despite the DSEC dataset capturing outdoor driving scenes with complex textures, motion patterns, and varying lighting conditions, the results in the table demonstrate that our method still exhibits impressive performance in optical flow estimation.
>
> | **Sequence**| &emsp;&emsp;&nbsp;&nbsp;&nbsp;**EPE**&emsp;&emsp; | &emsp;&emsp;&emsp;&nbsp;**Sequence**&emsp;&emsp; | &emsp;&emsp;&nbsp;&nbsp;&nbsp;**EPE**&emsp;&emsp; |
> |-|:-:|:-:|:-:|
> | *average*  | &emsp;&emsp; $6.649$ &emsp;&emsp; | &emsp;&emsp; *thun_01_b* &emsp;&emsp;|&emsp;&emsp; $6.925$ &emsp;&emsp;|
> | *interlaken_00_b*  | &emsp;&emsp; $12.000$ &emsp;&emsp; | &emsp;&emsp; *zurich_city_12_a* &emsp;&emsp;| &emsp;&emsp; $4.272$ &emsp;&emsp;|
> | *interlaken_01_a*  | &emsp;&emsp; $6.784$ &emsp;&emsp; | &emsp;&emsp; *zurich_city_14_c* &emsp;&emsp;| &emsp;&emsp; $7.604$ &emsp;&emsp; |
> | *thun_01_a*    | &emsp;&emsp; $5.209$ &emsp;&emsp; | &emsp;&emsp; *zurich_city_15_a* &emsp;&emsp;| &emsp;&emsp; $4.612$ &emsp;&emsp; |
>
> ---
>
> > **[W4, L1, L4] Static Background Assumption.**
>
> We have already discussed this scenario under **c) Dynamic scene** in **[W1, Q4, L5] Discussion about failure cases**. Please refer to that section.
>
> ---
>
> > **[Q1] Analysis of model capacity and motion decomposition strategy on performance.**
>
> The performance gain of MoFlow is not related to increased model capacity or the proposed motion decomposition strategy.
>
> **increased model capacity:** In our method, the Neural ODE is solely used to preserve the optical flow field; even a shallow MLP is sufficient to represent complex flow fields. Theoretically, increasing model capacity might yield slightly better fitting accuracy, but this improvement becomes negligible given the significant increase in memory and runtime. We tested the impact of model capacity on performance by adjusting the network depth of the MLP. In the original setting, the MLP network depth was 8, and we tested the performance with a network depth of 4. As shown in the results in the table below, the impact of model capacity on performance is minimal, which demonstrates that our method achieves strong performance without relying on simply increasing model capacity. In the table, the left column is $\text{EPE}$, the middle column is $RMS_{\omega}$, and the right column is $RMS_{v}$.
> | **Sequence**        | &emsp;&emsp;&emsp;&emsp;&emsp;**NN=4**&emsp;&emsp; | &emsp;&emsp;&emsp;&emsp;&emsp;**NN=8**&emsp;&emsp; |
> |-|:-:|:-:|
> | *indoor_flying1* | &emsp;&emsp; $1.52$ \| $3.40$ \| $0.11$ &emsp;&emsp; | &emsp;&nbsp;&nbsp;&nbsp;&nbsp; $1.56$ \| $3.44$ \| $0.11$ &emsp;&emsp; |
> | *indoor_flying2* | $2.06$ \| $5.21$ \| $0.13$ | $2.04$ \| $5.31$ \| $0.12$ |
> | *indoor_flying3* | $1.78$ \| $4.06$ \| $0.15$ | $1.84$ \| $4.12$ \| $0.15$ |
> | *outdoor_day1* | $1.71$ \| $3.44$ \| $0.86$ | $1.61$ \| $3.38$ \| $0.76$ |
> | *average* | $1.78$ \| $4.03$ \| $0.31$ | $1.76$ \| $4.06$ \| $0.29$ |
>
> **motion decomposition strategy:** As we just discussed, Neural ODEs and splines are merely representation tools. As long as their representational capacity is strong enough, adopting a motion decomposition strategy won't affect performance. While we could use a single network to represent both optical flow and motion simultaneously, we believe its performance would not surpass the current approach.
>
>
> [1] A Unifying Contrast Maximization Framework for Event Cameras, with Applications to Motion, Depth, and Optical Flow Estimation

---

> > ### Comment · Reviewer_f2hB · 2025-08-04
> >
> > Thank you for your response. I have a few follow-up questions:
> >
> > Do you have any quantitative comparison of inference time with other methods?
> >
> > Regarding the specific dataset where your method underperforms (as I mentioned earlier), should this be interpreted as a methodological weakness?
> >
> > Also, would it be possible for you to include a comparison table on the DSEC dataset against other methods? I believe explicitly showing this in the paper would significantly improve clarity and completeness.

---

> > > ### Author Response · Authors · 2025-08-09
> > > **Author Final Reply: Thanks for recognition and suggestions**
> > >
> > > We sincerely appreciate your insightful review and recommendation for acceptance.
> > >
> > > We are deeply grateful for your acknowledgment of our paper’s strengths, including its well-motivated problem setting, novel modular architecture, and self-supervised learning framework. We also sincerely appreciate your positive assessment for our strong experimental results, as well as the comprehensive ablation studies and visualizations, which effectively validate our design choices. Your recognition has been immensely encouraging.
> > >
> > > We particularly appreciate the productive discussions during the rebuttal phase regarding runtime efficiency, robustness in dynamic scenes, model capacity, and performance on more challenging datasets. Your constructive feedback has helped us refine these aspects significantly, and we will incorporate all relevant analyses into the final version to further strengthen the paper.
> > >
> > > Thank you again for your time and expertise in reviewing our work. We would be truly grateful if you could consider raising your rating based on our revisions and additional clarifications.

---

> > > > ### Comment · Reviewer_f2hB · 2025-08-09
> > > >
> > > > Thank you for your response. Most of my concerns have been resolved.

---

> ### Author Response · Authors · 2025-08-07
> **Discussion about  inference time and optical flow estimation on DSEC dataset**
>
> We sincerely appreciate the reviewer’s thoughtful comments and valuable suggestions, which have significantly helped us improve clarity of paper. Below, we address the reviewer’s questions in detail.
>
> To address the questions, we chose to use the DSEC dataset to evaluate the inference time and the optical flow estimation accuracy. The DSEC dataset is an outdoor autonomous driving dataset that includes diverse lighting conditions, fast motions, and complex scenes. We believe that this can fully demonstrate the performance of our method in challenging scenarios.
>
> We first present the average inference time of our method on the DSEC dataset and compare it with other methods. The results in the table indicate that our method achieves faster inference speed compared to most methods, demonstrating its efficiency.
> | **Method**        | &emsp;&emsp;&emsp;Inference Time&emsp;&emsp; |
> |-|:-:|
> | E-RAFT [1] | &emsp;&emsp; $46.33ms$ &emsp;&emsp; |
> | TamingCMax [2] | $40.10ms$ |
> | MotionPriorCMax [3] | $7.27ms$ |
> | MultiCMax [4] | $9.9 \times 10^3ms$ |
> | **Ours** | $30.15ms$ |
>
> Due to the large number of experiments and time constraints, we are still pushing the limit of our method's performance on the DSEC dataset. Our latest optical flow estimation results (EPE) are as follows. The results in the table show that our method is slightly inferior to other methods.
> | **Sequence**| &emsp;&emsp;&emsp;**Ours**&emsp;&emsp; | &emsp;&emsp;**MultiCMax [4]**&emsp;&emsp;|&emsp;&emsp;**MotionPriorCMax [3]**&emsp;&emsp;|&emsp;&emsp;**TamingCMax [2]** &emsp;&emsp;|
> |-|:-:|:-:|:-:|:-:|
> | *interlaken_00_b*  | &emsp;&emsp;&nbsp;&nbsp;&nbsp; $7.313$   &emsp;&emsp; | &emsp;&emsp; $5.744$ &emsp;&emsp;| &emsp;&emsp; $3.207$ &emsp;&emsp;|&emsp;&emsp; $3.337$ &emsp;&emsp;|
> | *interlaken_01_a*  | &nbsp;&nbsp;&nbsp;$3.549$  |  $3.743$ | $2.381$ | $2.489$ |
> | *thun_01_a*  | &nbsp;&nbsp;&nbsp;$3.601$  | $2.116$ |  $1.388$ | $1.730$ |
> | *thun_01_b*    | &nbsp;&nbsp;&nbsp;$2.987$ |  $2.480$ | $1.540$ | $1.657$ |
> | *zurich_city_12_a*    | &nbsp;&nbsp;&nbsp; $3.280$ | $3.862$ |  $8.325$ | $2.724$ |
> | *zurich_city_14_c*    | &nbsp;&nbsp;&nbsp; $4.433$ |  $2.724$ | $1.784$ | $2.635$ |
> | *zurich_city_15_a*    |&nbsp;&nbsp;&nbsp; $2.900$ |  $2.347$ | $1.454$ | $1.686$ |
> | *average*    | &nbsp;&nbsp;&nbsp; $3.828$ | $3.472$ | $3.195$ | $2.330$ |
>
> This demonstrates the potential of our method to achieve accurate optical flow estimation in challenging scenarios. It does not necessarily indicate a weakness in our methodology, but rather highlights areas where further refinement could enhance its performance.
>
> Once again, we thank the reviewer for constructive feedback and insightful questions. We believe these suggestions have greatly contributed to enhancing the clarity and completeness of our work.
>
> **Reference:**
>
> [1] E-RAFT: Dense Optical Flow from Event Cameras. (3DV'21)
>
> [2] Taming Contrast Maximization for Learning Sequential, Low-latency, Event-based Optical Flow. (ICCV'23)
>
> [3] Motion-prior Contrast Maximization for Dense Continuous-Time Motion Estimation. (ECCV'24)
>
> [4] Secrets of Event-Based Optical Flow. (ECCV'22)

---

### Official Review · Reviewer_3SAm · 2025-07-01

**Clarity:** 3
**Significance:** 2
**Originality:** 3
**Rating:** 4
**Confidence:** 4

**Summary:**

This paper introduces E-MoFlow, an unsupervised framework for jointly estimating optical flow and 6-DoF egomotion, which are typically addressed independently in 3D vision. The authors propose to overcome the ill-posed challenge of separate estimation, especially pertinent for neuromorphic vision without ground truth, by implicitly regularizing spatial-temporal and geometric coherence. E-MoFlow models camera egomotion as a continuous spline and optical flow as an implicit neural representation, integrating spatiotemporal coherence through inductive biases. Furthermore, it incorporates structure-and-motion priors via differential geometric constraints, avoiding explicit depth estimation while maintaining geometric consistency. Experiments demonstrate the framework's versatility and state-of-the-art performance in various 6-DoF motion scenarios under an unsupervised paradigm, even competing with supervised approaches.

**Questions:**

See weakness.

**Ethical Concerns:**

["NO or VERY MINOR ethics concerns only"]

**Final Justification:**

I keep my score.

**Limitations:**

See weakness.

**Quality:**

3

**Strengths And Weaknesses:**

The author's writing is clear, and the method is also easy to understand, specifically how the author optimizes optical flow and egomotion simultaneously in an unsupervised manner through events. However, I still have some doubts about the design of the loss function.

The design of the Differential Flow Loss uses the event's IWE from the CM algorithm as ground truth, then aligns it using the optical flow's ODE. Here I have some questions: If the supervisory signal comes from the CM algorithm, is the model's estimation upper limit determined by the CM algorithm (if not, please provide more explanation). What are the differences between this network-based estimation strategy and directly using the CM algorithm to estimate optical flow? Also, when the CM algorithm aligns events, it does so at time t and t_ref. How is it ensured that the network can learn continuous optical flow between these two moments? And how are the gradients for Equation (3) specifically calculated during backpropagation? Does it require multiple gradient accumulations for the flow model, and could this lead to potential OOM issues? Since I am not familiar with the gradient analysis for ODEs in Equations 3-5, I hope the author can provide more explanation in this part.

The ablation experiment shown in Tab.1 indicates that with the addition of unsupervised loss 'a', its performance is comparable to that with given gt motion. Has the model's robustness been tested here? Or are there any failure cases? Because the unsupervised loss designed by the author requires coupling between two losses, it certainly needs a more refined design compared to supervised methods. I am curious whether this algorithm can still achieve results comparable to supervised effects in cases of very fast motion.

Finally, the idea of jointly optimizing optical flow and egomotion is very interesting. If the authors are interested, they could try adding a multi-view constraint loss provided by 3DGS on top of this. This is quite similar to the geometric loss in the paper, but here it is afforded by 3DGS. Reference: USP-Gaussian: Unifying Spike-based Image Reconstruction, Pose Correction and Gaussian Splatting CVPR 2025

---

> ### Author Rebuttal · Authors · 2025-07-28
>
> ## Response to reviewer 3SAm
>
> We sincerely appreciate the reviewer’s recognition of our efforts in addressing the challenges of unsupervised optical flow and 6-DoF egomotion estimation through implicit neural representations and spatiotemporal coherence. Also, Thanks for the constructive and thoughtful comments.
>
> We address the main concerns below.
>
> ---
>
> > **[W1] Is CM algorithm the upper limit of E-MoFlow.**
>
> The performance of E-MoFlow is not determined by the upper limit of the CM algorithm. In contrast, our model uses two types of loss functions, which theoretically leads to better accuracy than relying solely on differential flow loss. This is further substantiated by the ablation studies presented in Table 1 of the main paper.
>
> The reason for this lies in the inherent limitations of a standalone differential flow loss, which, as illustrated in Figure 3 of our main paper, cannot provide enough constraints (known as the aperture problen or event collapse). Our differential geometric loss, however, offers additional implicit regularization for optical flow estimation, preventing the model from converging to an ill-posed local solution.
>
> ---
>
> > **[W2] Differences between the network-based estimation and directly using the CM algorithm.**
>
> Compared to traditional CM methods, our network-based estimation strategy leverages implicit neural representations to model optical flow, achieving implicit regularization of spatio-temporal flow continuity.
>
> Specifically, traditional non-learning CM algorithms [1] [2] typically model optical flow directly as a vector. This approach has several key limitations:
>
> 1. The optimized flow often struggles to effectively fit non-linear motion in complex scenarios.
>
> 2. The estimated flow only corresponds to the flow at the starting moment, making it impossible to obtain flow at arbitrary moments between the start and end timestamps. Consequently, it cannot estimate continuous-time flow.
>
> 3. Due to the absence of inherent implicit constraints on the flow, traditional CM often necessitates an additional loss function to enforce consistency between neighboring flow vectors.
>
> In contrast, our method models optical flow using implicit neural representations. This naturally introduces the Multi-Layer Perceptron (MLP) network's inherent bias towards learning low-frequency signals, which effectively constrains the spatio-temporal continuity of the estimated flow.
>
> ---
>
> > **[W3] Can E-MoFlow learn continuous optical flow.**
>
> It is ensured that our method can learn continuous optical flow. Specifically, there are two main reasons for this:
>
> 1. Our reference time is randomly selected in each iteration. This effectively prevents the degradation that can occur when only a fixed reference time is chosen.
>
> 2. This is also attributed to our use of Neural Ordinary Differential Equations (Neural ODEs) to parameterize the optical flow. This enables us to obtain optical flow at any given moment and to acquire displacement over any time interval through integration.
>
> ---
>
> > **[W4] Details of the gradient calculation.**
>
> During backpropagation, the gradients for Equation (3) do not require multiple gradient accumulations for the flow model, and do not lead to potential OOM issues. Below are the specific reasons and analysis.
>
> Equation (2) represents a neural ODE system, which parameterizes the optical flow at different times and states using a neural network. Equation (3) illustrates how to calculate the displacement from time $t_k$ to time t by forward-integrating the neural ODE-parameterized flow. Equation (4) is the adjoint ODE system of the neural ODE system. This is a backward system, which can be understood as the sensitivity of the loss with respect to the state x. When Equation (4) is combined with the partial derivatives of the neural network's output with respect to its parameters, we can obtain the partial derivatives of the loss with respect to the neural network parameters at each moment. At this point, Equation (5) shows that to get the derivative of the loss with respect to the neural network parameters along the entire path, we only need to integrate backward to the starting point.
>
> This process is fundamentally no different from backpropagation; in fact, you can prove they are exactly the same. The only distinction is that, compared to the "discretize then differentiate" approach of classical backpropagation, the adjoint ODE method follows a "differentiate then discretize" strategy. It transforms backpropagation into an ODE integration, thereby avoiding the storage of large quantities of intermediate variables and complex gradient computation graphs. This characteristic prevents the problems of gradient explosion and excessive memory usage often encountered when applying classical backpropagation to neural ODEs.
>
> ---
>
> > **[W5] Robustness to very fast motion.**
>
> Our method remains robust and demonstrates good performance even in very fast motion scenarios. Thanks to our continuously parameterized optical flow and motion, our method can theoretically handle fast, non-linear movements without any performance loss. We conducted experiments on the *outdoor_day1* sequence to further verify this. The *outdoor_day1* sequence in the MVSEC dataset was captured by a vehicle-mounted event camera, with camera motion speeds reaching 12m/s. The results in the table below validate the robustness of our method in very fast motion scenarios.
> | **Sequence**        | &emsp;&emsp;$\mathrm{EPE}$&emsp;&emsp; | &emsp;&emsp; $\mathrm{RMS}_{\omega}$ &emsp;&emsp; | &emsp;&emsp; $\mathrm{RMS}_{v}$&emsp;&emsp; |
> |-|:-:|:-:|:-:|
> | *outdoor_day1* | &emsp;&emsp; $1.61$ &emsp;&emsp; | &emsp;&emsp; $3.38$ &emsp;&emsp; | &emsp;&emsp; $0.76$ &emsp;&emsp; |
>
> ---
>
> > **[W6] More discussions on related works.**
>
> We will expand our discussion to include 3D Gaussian Splatting (3DGS). Our method utilizes a differential geometric error, which shares inspiration with the multi-view geometric error used in 3DGS. In future work, we will intend to explore combining these two loss functions.
>
> ---
>
> **Reference:**
>
> [1] A Unifying Contrast Maximization Framework for Event Cameras,
> with Applications to Motion, Depth, and Optical Flow Estimation
>
> [2] Focus Is All You Need: Loss Functions For Event-based Vision

---

> > ### Comment · Reviewer_3SAm · 2025-08-06
> > **Thanks for solving my problem**
> >
> > I have read the rebuttal, and thanks to the authors for solving my problems.

---

> > > ### Author Response · Authors · 2025-08-07
> > > **Author Final Reply: Thanks for recognition and suggestions**
> > >
> > > We sincerely thank you for your detailed and thoughtful review, as well as for recommending our paper for acceptance.
> > >
> > > We deeply appreciate your recognition of our efforts in exploring how to jointly learn optical flow and 6-DoF motion using an unsupervised learning paradigm. Your insightful discussion of our methodological contributions, particularly the integration of implicit neural representations and neural ODEs to enforce spatiotemporal continuity constraints on optical flow, is highly encouraging.
> > >
> > > We are pleased to have addressed your concerns comprehensively and would be grateful if you could kindly consider raising the rating. The relevant results will be included in the final version of the paper.

---

### Official Review · Reviewer_j1WT · 2025-07-08

**Clarity:** 2
**Significance:** 3
**Originality:** 3
**Rating:** 5
**Confidence:** 4

**Summary:**

E-MoFlow presents a novel approach to the problem of jointly predicting optical flow and ego-motion from sequential data obtained from an event camera. It's main contribution is an implicit neural flow field formulation which can be optimized at test time and solved integrated like an ODE to produce a continuous flow fields. In addition to optimizing the neural flow field, a cubic spline is also fit for the camera's motion. Both the spline and the flow field are optimized together in order to produce the final output, and they are trained using a combination of two losses. One is a flow specific loss which uses contrast maximization (as in prior work) and the other is a differentiable geometric loss which connects the flow and ego motion together as is commonly done in multi-view geometry.

**Questions:**

1. In table 3, why are some of the methods missing for certain scenes? Is this because they were using those scenes for training?
2. Since your method does not crop inputs, is this an advantage in the qualitative results? Why do other methods use a crop? Runtime issues?
3. Have you tried early-stopping instead of training every event for 1000 steps? In related works (see below) early stopping can improve runtime performance quite significantly.

I think this paper would be well served by making more parallels to other related literature that also does neural representations and optimization. In particular the NeRF literature [1], as well as the related work in scene flow such as Neural Scene Flow Prior [2], Neural Trajectory Prior [3], and Neural Eulerian Scene Flow Fields [4].

On the topic of the related works, in Neural Eulerian Scene Flow Fields it was shown that by performing Euler integration over the neural ODE representation, continuous point tracking was an emergent property. I imagine that E-MoFlow could exhibit similar properties and this would be a compelling visual to add (even if no metrics for point tracking can be provided).

1. https://arxiv.org/abs/2003.08934
2. https://arxiv.org/abs/2111.01253
3. https://openaccess.thecvf.com/content/CVPR2022/papers/Wang_Neural_Prior_for_Trajectory_Estimation_CVPR_2022_paper.pdf
4. https://arxiv.org/abs/2410.02031

**Ethical Concerns:**

["NO or VERY MINOR ethics concerns only"]

**Final Justification:**

The authors have satisfactorily answered my questions, and will include the relevant parallels to related work in their camera ready. Their method benefits from the same tricks as many other neural ODE solvers for flow methods such as early stopping, and continuous point tracking behavior, and they included these results in the rebuttal. My score is now accept.

**Limitations:**

yes

**Paper Formatting Concerns:**

Minor nit:
Table 1 caption has a typo: "Ablation studies on differential geometric constrain" should read "constraints" at the end of this sentence.

**Quality:**

3

**Strengths And Weaknesses:**

## Strengths
The main strengths of this method is that it is fully unsupervised and does not require depth priors, while still producing optical flow estimates that are quite good. In certain circumstances, other methods may perform better than E-MoFlow through the use of depth or supervised learning. The overall method is relatively well explained, though a pseudocode section to explain the optimization loop would probably be helpful.

## Weaknesses
This paper's main weaknesses are in its presentation. The method could use some extra explanation (it's never explicitly stated in the paper that it performs test time optimization). Similarly, the runtime is not stated which is a notable con of test time optimization approaches.

---

> ### Author Rebuttal · Authors · 2025-07-28
>
> ## Response to Reviewer j1WT
>
> We sincerely thank the reviewer for recognizing the key strengths of our method, including its fully unsupervised nature and its ability to produce competitive optical flow estimates without requiring depth priors. We also appreciate the valuable suggestions, which we will incorporate to improve our method.
>
> We address the main concerns below.
>
> ---
>
> > **[W1] Lack of explicit explanation that the paper performs test time optimization.**
>
> In the camera-ready version, we'll explicitly explain that our method is a test-time optimization approach.
>
> ---
>
> > **[W2] Lack of runtime analysis.**
>
> We report the training time and the inference time of our method in the following table. The analysis has been tested across all sequences in dataset MVSEC. Each sequence is divided into multiple clips with a time interval of approximately 80-100 ms, corresponding to $dt=4$, and each clip contains approximately 30,000 event data. In the table below, we report the average training time and inference time per clip.
> | **Sequence**| &emsp;&emsp;**Training Time**&emsp;&emsp; | &emsp;&emsp;**Inference Time**&emsp;&emsp; |
> |-|:-:|:-:|
> | *indoor_flying1*  | &emsp;&emsp; $9.30s$ &emsp;&emsp; | &emsp;&emsp; $0.86s$ &emsp;&emsp;|
> | *indoor_flying2*  | &emsp;&emsp; $9.41s$ &emsp;&emsp; | &emsp;&emsp; $0.87s$ &emsp;&emsp;|
> | *indoor_flying3*  | &emsp;&emsp; $9.50s$ &emsp;&emsp; | &emsp;&emsp; $0.88s$ &emsp;&emsp;|
> | *outdoor_day1*    | &emsp;&emsp; $10.77s$ &emsp;&emsp; | &emsp;&emsp; $0.86s$ &emsp;&emsp;|
>
> It's important to note that these runtime results are preliminary, as our method doesn't employ any acceleration techniques, and a simple acceleration strategy (such as the early-stopping evaluated in Q3) could significantly accelerate the method. We will include the detailed runtime analysis in the camera-ready version.
>
> ---
>
> > **[W3] Runtime is a notable con of test time optimization approaches.**
>
> We acknowledge that runtime is a drawback of test-time optimization approaches. However, we'd like to point out that in future work, we'll try using the proposed implicit regularization to unsupervisedly train a feed-forward optical flow prediction network. At that point, its runtime will no longer be a limitation.
>
> ---
>
> >  **[Q1] Explanation for some missing scenes in table 3.**
>
> AEmin [1], IncEmin [2], and PEME [3] methods fail on these scenes and ECN [4] struggles to generalize well in complex indoor scene. The detailed analysis is as follows.
>
> **AEmin [1], IncEmin [2], and PEME [3]** methods for estimating 6-DOF motion all require ground-truth depth measurements to augment the event data. However, the depth values obtained from LiDAR are measured at a frequency of only 20 fps, which is much lower than the temporal resolution of the event data. This discrepancy introduces significant noise into the depth-augmented event data. The *indoor_flying2* and *indoor_flying3* sequences feature complex motion patterns and rich scene texture information, which make these methods less robust to noise, leading to their failure on these sequences.
>
> **ECN [4]** requires training on a large amount of event data, but the indoor scene sequences in MVSEC are too short to generate sufficient training data and contain more complex motion patterns. As mentioned in the ECN paper [4], due to the limited training data, ECN struggles to generalize well in complex indoor scenes. Furthermore, its strong performance in outdoor scenes can be attributed to the fact that only two motion parameters play a critical role (forward translation and yaw axis rotation in the vehicle coordinate frame), leading the network to tend to overfit.
>
> ---
>
> >  **[Q2] Explanation for no image crop.**
>
> Our method does not require cropping the input, which indeed provides an advantage in the qualitative results. Other methods use cropping primarily due to limitations of their approach. The detailed analysis is as follows.
>
> **a)**  Our model uses a coordinate-MLP to track the motion of events in the $(x, y, t)$ space. We aim for the coordinate-MLP to fully utilize the information from all event data to learn continuous-time motion in the spatio-temporal domain, while leveraging the implicit regularization of neural ODEs to constrain the motion smoothness of adjacent events. Therefore, it is crucial to avoid cropping, which could result in the loss of local spatio-temporal information and disrupt the implicit regularization of motion patterns. This allows us to recover optical flow at the original resolution using the raw event data.
>
> **b)** Other learning-based methods, such as EV-FlowNet [5] and ConvGRU-EV-FlowNet [6], require preprocessing the raw event data into event images as inputs to their networks. Due to the spatial sparsity of event data, event images may contain a large number of regions where no events occur, which contribute little to optical flow estimation. As a result, these methods often use cropping to focus primarily on regions with dense events. In contrast, our method does not require preprocessing the raw events into event images. Instead, it directly uses the $(x, y, t)$ coordinates of the raw events obtained from actual measurements to train the coordinate-MLP. Our method does not introduce a large number of coordinates from regions without events, which lack effective information, for training. Instead, it naturally supports focusing only on regions where real events are triggered. Additionally, the CNN employed by these methods require input data with a fixed resolution, and cropping the event images to a fixed size simplifies the design and training of the model.
>
> **c)** Cropping the input primarily affects the size of data ultimately fed into the network for training, which significantly impacts memory usage, especially with high-resolution inputs. However, for runtime performance, since neural network training and inference leverage the parallel computing power of GPUs, cropping does not have a significant impact on runtime.
>
> ---
>
> >  **[Q3] Results of early stop training strategy.**
>
> We adopt the early-stopping strategy and report the detailed results in the following tables. We find that it significantly reduces the training time with only a slight loss in accuracy, achieving an excellent trade-off. Specifically, we set the patience to 45 and the minimum improvement threshold to 1.0e-3, applying the early stopping strategy after 300 iterations. We find that the training speed was boosted by 2.2x, while the optical flow estimation accuracy showed only a slight 3.41% drop in the EPE. The motion estimation accuracy still maintain SOTA performance. The detailed results are shown in the table below.
>
> **a)** `Training time.` The number in brackets indicates the average training iterations when using early-stopping strategy.
> | **Sequence**        | &emsp;**w/ early stopping**&emsp; | &emsp;**w/o early stopping**&emsp; | &emsp;**Efficiency Improvement**&emsp; |
> |-|:-:|:-:|:-:|
> | *indoor_flying1* | &emsp;&emsp; $4.21s$ $(452.93)$ &emsp;&emsp; | &emsp;&emsp; $9.30s$ &emsp;&emsp; | &emsp;&emsp; $2.21 \times$ $\uparrow$ &emsp;&emsp; |
> | *indoor_flying2* | $4.30s$ $(461.84)$ | $9.41s$ | $2.19 \times$ $\uparrow$ |
> | *indoor_flying3* | $4.78s$ $(496.77)$ | $9.50s$ | $1.99 \times$ $\uparrow$ |
> | *outdoor_day1* | $4.56s$ $(487.51)$ | $10.77s$ | $2.36 \times$ $\uparrow$ |
> | *average* | $4.46s$ $(474.76)$ | $9.75s$ | $2.19 \times$ $\uparrow$ |
>
> **b)** `Optical flow estimation accuracy(EPE).`
> | **Sequence**        | &emsp;**w/ early stopping**&emsp; | &emsp;**w/o early stopping**&emsp; | &emsp;**Performance Change**&emsp; |
> |-|:-:|:-:|:-:|
> | *indoor_flying1* | &emsp;&emsp; $1.61$ &emsp;&emsp; | &emsp;&emsp; $1.56$ &emsp;&emsp; | &emsp;&emsp; $3.21$ % $\downarrow$ &emsp;&emsp; |
> | *indoor_flying2* | $2.09$ | $2.04$ | $2.45$ % $\downarrow$ |
> | *indoor_flying3* | $1.90$ | $1.84$ | $3.26$ % $\downarrow$ |
> | *outdoor_day1* | $1.68$ | $1.61$ | $4.35$ % $\downarrow$ |
> | *average* | $1.82$ | $1.76$ | $3.41$ % $\downarrow$ |
>
> **c)** `6-DoF egomotion estimation.` Each column reports RMSw on the left and RMSv on the right.
> | **Sequence**        | &emsp;**w/ early stopping**&emsp; | &emsp;**w/o early stopping**&emsp; |
> |-|:-:|:-:|
> | *indoor_flying1* | &emsp;&emsp; $5.32$ \| $0.12$ &emsp;&emsp; | &emsp;&emsp; $3.44$ \| $0.11$ &emsp;&emsp; |
> | *indoor_flying2* | $5.77$ \| $0.14$ | $5.31$ \| $0.12$ |
> | *indoor_flying3* | $4.94$ \| $0.17$ | $4.12$ \| $0.15$ |
> | *outdoor_day1* | $5.22$ \| $1.41$ | $3.38$ \| $0.76$ |
>
> ---
>
> >  **[Q4] More discussions on neural representations and optimization.**
>
> We will include discussions of these related works in implicit neural flow representation in our camera-ready version.  Our work, which focuses on representing continuous flow in the $(x,y,t)$ spatiotemporal domain, shares a very similar inspiration with these works that model object motion and deformation in 3D space. Both aim to implicitly constrain the smoothness of continuous-time flow through neural representation priors, while modeling continuous trajectories via ODEs or interpolation.
>
> ---
>
> >  **[Q5] More continuous point tracking visual results.**
>
> We will include visualized results of point tracking in the camera-ready version. Due to the rebuttal policy prohibiting image uploads, we're unable to show you these visual results.
>
> ---
>
> >  **[MW1] Typo.**
>
> We will correct this typo in the future version.
>
> ---
>
> **Reference:**
>
> [1] Entropy Minimisation Framework for Event-based Vision Model Estimation
>
> [2] Robust Event-based Vision Model Estimation by Dispersion Minimisation
>
> [3] Progressive Spatio-temporal Alignment for Efficient Event-based Motion Estimation
>
> [4] Unsupervised Learning of Dense Optical Flow, Depth and Egomotion from Sparse Event Data
>
> [5] EV-FlowNet: Self-Supervised Optical Flow Estimation for Event-based Cameras
>
> [6] Self-Supervised Learning of Event-Based Optical Flow with Spiking Neural Networks

---

> > ### Comment · Reviewer_j1WT · 2025-08-05
> >
> > Fantastic work! I'm glad to see that the early stopping was a useful trick carried over from the other neural prior based scene flow methods, and that it made a positive impact on the runtime. I'm looking forward to some nice point tracks in the updated manuscript as well. All my questions have been answered :)

---

> > > ### Author Response · Authors · 2025-08-05
> > >
> > > Thank you very much for your recognition of our work and constructive suggestions on the method. We will incorporate those results and additional clarifications into the final version of our paper.

---

### Note · Authors · 2025-08-16

We sincerely thank all reviewers for their thoughtful review and constructive feedback. We also deeply appreciate their recognition and support for our work. As highlighted, our approach offers a novel unsupervised framework that simultaneously learns optical flow and egomotion without relying on depth priors, while still achieving competitive performance. We are especially grateful for the acknowledgment of our novel approach that leverages implicit neural representations and Neural ODEs to enforce spatiotemporal smoothness in optical flow estimation, while introducing differential geometric loss to implicitly maintain geometric consistency between optical flow and camera motion. The effectiveness of these methodological innovations has been convincingly demonstrated through the method's competitive performance in both optical flow estimation and 6-DoF motion estimation tasks.

During the rebuttal and discussion phases, we have primarily addressed several key concerns raised by the reviewers and incorporated their constructive suggestions.
- We conducted a runtime analysis of Emo-Flow and further experimented with an early-stopping training strategy, which was found to boost training speed by 2.2×.
- We have provided a detailed explanation of how Emo-Flow employs implicit neural networks and Neural ODEs to learn continuous optical flow, and highlighting the differences from the traditional CMax algorithm. It further elaborates on how gradients are backpropagated during the training of Neural ODEs.
- We investigated the robustness of Emo-Flow under fast motion, varying illumination, pure rotation, and dynamic scenes. Experiments demonstrate that our method is robust to fast motion and pure rotation scenarios, while the introduction of the Huber loss further improves its robustness in dynamic scenes.
- We tested the performance of Emo-Flow on the more challenging autonomous driving dataset DSEC, and the results demonstrate its potential to achieve accurate optical flow estimation in complex scenarios.
- We demonstrate that MoFlow achieves strong performance without relying on merely increasing model capacity, which primarily stems from the powerful representational capabilities of Neural ODEs and splines in modeling continuous optical flow and camera motion.

We will incorporate the discussion and results from the rebuttal phase into the final version of our paper. Once again, we sincerely appreciate the reviewers for their thorough and insightful feedback!

---

### Decision · Program_Chairs · 2025-09-17

**Decision:**

Accept (poster)

**Comment:**

The paper proposes a method to estimate egomotion and optical flow jointly from event data. The method is unsupervised, an important feature as large event datasets are infeasible to collect. The reviewers appreciated the problem setting and the method. The main weaknesses in the reviews were related to missing evaluations (e.g., highly dynamic scenes and ablations). The rebuttal addressed these concerns and presented several new results. All reviewers were satisfied with the additional experiments and recommended accepting the paper.  I agree--the paper provides useful contributions to perception from event stream data.